# Enhancing 3D Reconstruction for Dynamic Scenes

**Jisang Han[1*] Honggyu An[1*] Jaewoo Jung[1*] Takuya Narihira[2] Junyoung Seo[1]**
**Kazumi Fukuda[2] Chaehyun Kim[1] Sunghwan Hong[3] Yuki Mitsufuji[2,4†] Seungryong Kim[1†]**

[1] KAIST AI    [2] Sony AI    [3] ETH Zürich AI Center, CVG, PRS    [4] Sony Group Corporation

https://cvlab-kaist.github.io/DDUSt3R

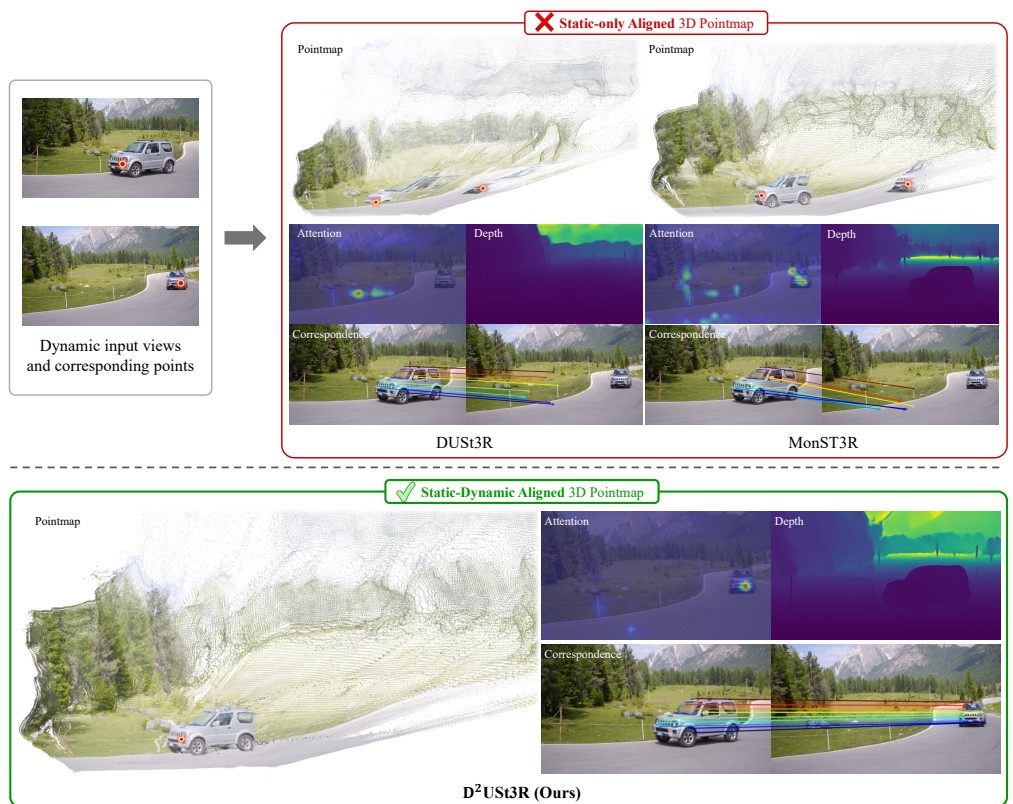

Figure 1: **Teaser.** Given a pair of input views, **D²USt3R** accurately establishes dense correspondence not only in static regions but also in dynamic regions, and enables 3D reconstruction of a dynamic scene via our proposed static-dynamic aligned pointmap. The colored ● pointmaps highlight that DUSt3R [44] and MonST3R [51] align pointmaps solely based on camera motion, causing corresponding 2D pixels within dynamic object misaligned. We also compare the cross-attention maps, established correspondence fields, and estimated depth maps produced by **D²USt3R** against baseline methods [44, 51], where our method shows higher precision.

## Abstract

In this work, we address the task of 3D reconstruction in dynamic scenes, where object motions frequently degrade the quality of previous 3D pointmap regression methods, such as DUSt3R, that are originally designed for static 3D scene reconstruction. Although these methods provide an elegant and powerful solution in static settings, they struggle in the presence of dynamic motions that disrupt alignment based solely on camera poses. To overcome this, we propose **D²USt3R** that directly regresses Static-Dynamic Aligned Pointmaps (SDAP) that simultaneiously

39th Conference on Neural Information Processing Systems (NeurIPS 2025).

capture both static and dynamic 3D scene geometry. By explicitly incorporating both spatial and temporal aspects, our approach successfully encapsulates 3D dense correspondence to the proposed pointmaps, enhancing downstream tasks. Extensive experimental evaluations demonstrate that our proposed approach consistently achieves superior 3D reconstruction performance across various datasets featuring complex motions.

# 1 Introduction

Recovering 3D scene geometry from images remains a core problem in computer vision. Traditional approaches, such as Structure-from-Motion (SfM) [33] and Multi-View Stereo (MVS) [35], have achieved impressive results in this context. While these methods are originally designed for recovering precise 3D scene geometry, they often struggle with scenes that include objects, symmetries, imagery with minimal overlapping and textureless regions [26, 34, 3, 9].

To overcome these limitations, recent approaches [44, 21, 41] leverage a learning framework to streamline the 3D reconstruction pipeline and enhance performance. DUSt3R [44], as a pioneering method, introduced a unified learning-based framework for dense stereo 3D reconstruction. Specifically, DUSt3R directly regresses 3D pointmaps that encode scene geometry, pixel-to-scene correspondences, and inter-view relationships, mitigating error accumulation typical in multi-stage pipelines. Despite its strengths in static scenarios, DUSt3R significantly struggles with dynamic scenes due to its rigidity assumption, as exemplified in Figure 1.

Dynamic scenes, prevalent in real-world scenarios, pose significant challenges in 3D scene reconstruction task, as object motions disrupts the camera pose-based alignment [44], causing misaligned correspondences and inaccurate depth estimates in dynamic objects, which further degrades reconstruction accuracy in static regions. While recent methods such as MonST3R [51] extends training to dynamic-scene video collections to account for dynamic objects, it still models all pointmaps as if generated by a single global rigid transformation. Consequently, these approaches suffer from compromised correspondence learning for dynamic objects, in turn impairing depth accuracy and robust geometry recovery.

In this paper, we propose **Dynamic Dense Unconstrained Stereo 3D Reconstruction (D$^2$USt3R)**, a novel feed-forward framework that directly regresses Static-Dynamic Aligned Pointmaps (SDAP), simultaneously accounting for both spatial structures and temporal motions to enable more reliable 3D reconstruction of both static and dynamic regions. Unlike MonST3R [51], which overlooks correspondences on moving objects, our model captures dense inter-frame matches by treating correspondence and reconstruction as a unified problem in dynamic scenes. We achieve this with a novel training scheme that applies separate supervisory signals to static and dynamic regions, signals that are further stabilized and localized to regions of interests by our occlusion and dynamic masks. Experimental results and visual comparisons confirm that our approach delivers a significant boost in reconstruction accuracy.

Our contributions are summarized as follows:

- Our approach, **D$^2$USt3R**, captures dynamic motion by leveraging static–dynamic aligned pointmaps, allowing for the comprehensive 3D reconstruction of all scene elements in any environment.
- To compensate for the missing direct 3D correspondences between dynamic objects, we propose a 3D alignment loss that effectively accounts for the occlusions and object motions.
- **D$^2$USt3R** achieves state-of-the-art performance across several downstream tasks, including multi-frame depth estimation as well as camera pose estimation, demonstrating superior results in dense 3D reconstruction of dynamic scenes.

# 2 Related Work

**Per-scene 3D reconstruction.** Classical 3D reconstruction methods typically follow a multi-stage pipeline to recover scene geometry and camera parameters from a set of uncalibrated images. Prominent examples include Structure-from-Motion (SfM) [52], and Simultaneous Localization and

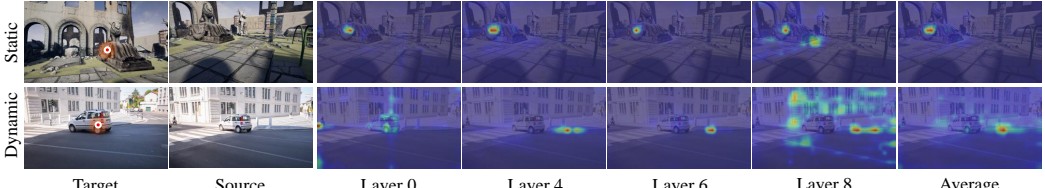

| | Target | Source | Layer 0 | Layer 4 | Layer 6 | Layer 8 | Average |

Figure 2: **Cross-attention visualization of DUSt3R [44] on static vs. dynamic scenes.** We show source-image attention maps for a highlighted query point (red) at each layer and averaged across layers. While DUSt3R captures geometric correspondences well in static regions, it fails in dynamic areas due to its rigid-motion, static-frame assumption.

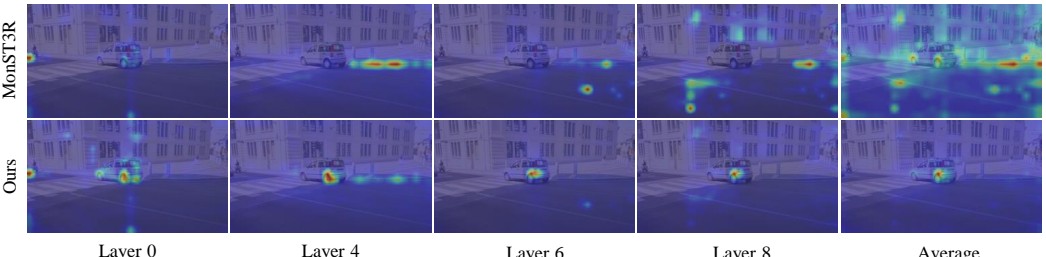

| Layer 0 | Layer 4 | Layer 6 | Layer 8 | Average |

Figure 3: **Cross-attention visualization on dynamic scenes: MonST3R [51] vs. Ours.** Using the same setup as Figure 2, MonST3R inherits DUSt3R's static-scene supervision and fails to align moving regions, limiting reconstruction. In contrast, our method consistently matches dynamic frames and produces sharply localized attention.

Mapping (SLAM) [8]. SfM incrementally reconstructs sparse 3D points through feature matching and bundle adjustment where SLAM simultaneously estimates camera trajectories and builds sparse or semi-dense maps in real-time. Building upon SfM, ParticleSfM [53] incorporate particle-based trajectory modeling to track object motions, paving the way for reconstruction in dynamic scenes. Despite all of these approaches showing remarkable performance, they typically require long processing time and resources for scene-specific optimization and often struggle with error accumulations from multi-stage pipelines.

**Learning-based static scene reconstruction.** Building on top of estabilished correspondence fields [4, 5, 17, 18, 14], various learning-based approaches [41, 25, 12, 37, 16] have been proposed to reconstruct static 3D scenes by learning strong 3D priors, representing the scene as point clouds [25, 13], meshes [12, 42], voxels [37, 6] and 3DGS [15]. Recently, DUSt3R [44] notably provides a unified, feed-forward pipeline for dense stereo matching, geometry estimation, and triangulation by directly regressing structured 3D pointmaps. Although DUSt3R significantly improves reconstruction quality and efficiency by reducing cumulative errors, DUSt3R is inherently designed for static scenes, limiting its effectiveness in scenarios involving dynamic components.

**Learning-based dynamic scene reconstruction.** Dynamic scene reconstruction introduces additional complexities due to non-rigid transformations occurring across frames. Similarly to static scene reconstruction, several recent approaches [23, 51, 28, 24, 43, 48] have employed learning-based methods to tackle dynamic scene reconstruction. Among these, MonST3R [51] directly fine-tunes DUSt3R using videos consisting of dynamic scenes. Although it has shown competitive performance, MonST3R retains DUSt3R's per-frame training paradigm and lacks explicit mechanisms for linking corresponding points across frames in dynamic scenes, leading to inconsistent depth estimations that arises from lacking constraints that fail to capture the intricate motion patterns of objects. Our method addresses this limitation by augmenting the existing feed-forward framework with motion-aware training objectives, explicitly enforcing consistent 3D point correspondences over time.

## 3 Preliminary

Given a pair of input images $I^1, I^2 \in \mathbb{R}^{W \times H \times 3}$, DUSt3R [44] predicts a pair of 3D pointmaps $X^{1,1}, X^{2,1} \in \mathbb{R}^{W \times H \times 3}$ for both images, each expressed in the camera coordinate system of $I^1$. To train the network in a supervised manner, ground-truth pointmaps for each image are defined in the coordinate space of the first camera. Specifically, given the camera intrinsics matrix $K \in \mathbb{R}^{3 \times 3}$,

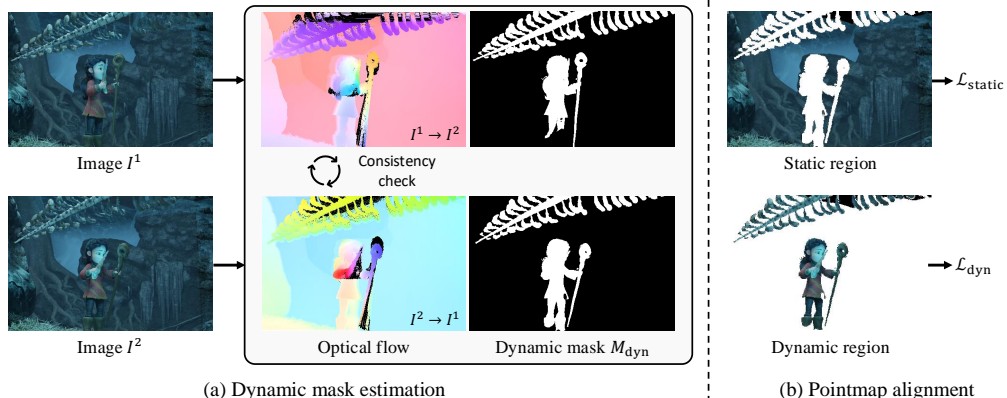

(a) Dynamic mask estimation      (b) Pointmap alignment

Figure 4: **Construction of alignment loss defined at static and dynamic regions.** We introduce a pipeline for constructing a static-dynamic aligned pointmap (SDAP) that explicitly handles occlusions in dynamic regions. First, we compute and refine optical flow via cycle consistency checks and derive a dynamic mask $M_{\text{dyn}}$. To align image $I^2$ with image $I^1$, we then: 1) warp static pixels using the known camera poses, and 2) warp dynamic pixels using the optical flow. By combining these two warps, we finally obtain SDAP that registers every corresponding 2D pixel into 3D space.

world-to-camera pose matrices $P^n, P^m \in \mathbb{R}^{4 \times 4}$ for images $n$ and $m$, and a ground-truth depth map $D \in \mathbb{R}^{W \times H}$, the ground-truth pointmap is computed as $\bar{X}^{n,m} = P^m(P^n)^{-1}h(K^{-1}D)$, where $h : (x, y, z) \mapsto (x, y, z, 1)$ represents the transformation to homogeneous coordinates.

Using 3D pointmaps, DUSt3R learns its parameters by minimizing the Euclidean distance between the ground-truth pointmaps $\bar{X}^{1,1}, \bar{X}^{2,1}$ and the predicted pointmaps $X^{1,1}, X^{2,1}$ for two corresponding sets of valid pixels $\mathcal{D}^1, \mathcal{D}^2 \subseteq \{1 \ldots W\} \times \{1 \ldots H\}$ on which ground-truth defined using a regression loss defined as:

$$\mathcal{L}_{\text{regr}}(v, i) = \left\| \frac{1}{z} X_i^{v,1} - \frac{1}{\bar{z}} \bar{X}_i^{v,1} \right\|, \tag{1}$$

where $v \in \{1, 2\}$ is the input views and $i \in \mathcal{D}^v$ denotes valid pixel positions. The scaling factors $z = \text{norm}(X^{1,1}, X^{2,1})$ and $\bar{z} = \text{norm}(\bar{X}^{1,1}, \bar{X}^{2,1})$ are computed using the normalization function:

$$\text{norm}(X^1, X^2) = \frac{1}{|\mathcal{D}^1| + |\mathcal{D}^2|} \sum_{v \in \{1,2\}} \sum_{i \in \mathcal{D}^v} \|X_i^v\|. \tag{2}$$

Additionally, DUSt3R incorporates a confidence score to learn to reject errorneously defined GT pointmaps, and it is included in the final loss function such that:

$$\mathcal{L}_{\text{conf}} = \sum_{v \in \{1,2\}} \sum_{i \in \mathcal{D}^v} C_i^{v,1} \mathcal{L}_{\text{regr}}(v, i) - \alpha \log C_i^{v,1}. \tag{3}$$

## 4 Methodology

### 4.1 Motivation and overview

As shown in Figure 1, DUSt3R [44] predicts 3D pointmaps in static regions with high accuracy, leveraging precise stereo correspondences for robust reconstruction. Inspired by ZeroCo's finding that DUSt3R's cross-attention inherently encodes geometric correspondences [1], we visualize attention maps for both static and dynamic scenes in Figure 2. While DUSt3R excels in static areas, it fails to produce localized attention scores in dynamic regions, leading to noisy pointmap predictions as visualized in Figure 1. To address this, MonST3R [51] augment DUSt3R's training data with dynamic sequences, but they still neglect explicit object-to-object correspondences. This omission prevents dynamic objects from serving as anchors that could strengthen the spatial structure and improve depth estimates of neighboring static regions, thereby limiting overall reconstruction quality. Figure 3 highlights these shortcomings, showing how noisy or missing correspondences degrade accuracy.

To overcome these limitations, we propose Static–Dynamic Aligned Pointmaps (SDAP). Unlike static-only aligned pointmaps, which align 3D points solely with rigid transformations, SDAP

Table 1: **Training datasets.** All datasets consists of synthetic scenes and provide both camera and depth. We excluded scenes containing incomplete dataset annotations or noisy objects (e.g., smoke). Details regarding each scene are documented in the supplementary material.

| Dataset | Domain | # of frames | # of Scenes | Dynamics | Dynamic mask | Optical flow | Ratio |
|---|---|---|---|---|---|---|---|
| Blinkvision Outdoor [22] | Outdoors | 6k | 23 | Realistic | ✓ | ✓ | 38.75% |
| Blinkvision Indoor [22] | Indoors | 6k | 24 | Realistic | ✓ | ✓ | 23.75% |
| PointOdyssey [54] | Indoors & Outdoors | 200k | 131 | Realistic | ✗ | ✗ | 12.5% |
| TartanAir [45] | Indoors & Outdoors | 1000k | 163 | None | ✗ | ✓ | 12.5% |
| Spring [29] | Outdoors | 6k | 37 | Realistic | ✓ | ✓ | 12.5% |

simultaneously aligns both static and dynamic components in the scene, fully encoding each pixel's spatial and temporal information. In the following sections, we describe our proposed SDAP representation and outline our training procedure.

## 4.2 Static-Dynamic Aligned Pointmap

While 3D pointmaps excel at encoding static 3D structure [44], they break down on moving objects due to temporal misalignment. To address this, we introduce Static–Dynamic Aligned Pointmaps (SDAP), which jointly enforce spatial consistency and capture pixel motion across time. Our key insight is to represent every pixel in a unified world coordinate system that aligns its position at each timestep, yielding a cohesive reconstruction of dynamic scenes. However, dynamic motion often leads to occlusions, meaning some pixels appear in only one frame. This results in incomplete alignment, which ultimately limits the quality of 3D reconstruction.

**Occlusion Masks.** To eliminate occluded regions in our SDAP representation, we first obtain dense 2D correspondences across each image pair using an off-the-shelf optical flow estimator [46]. If ground-truth flows are available in the dataset, we use those instead. In scenarios with large camera baselines, where occlusions become more frequent, we further apply a forward–backward consistency check [39, 50] to derive precise occlusion masks. The full procedure is detailed below.:

$$p_2' = p_1 + \mathbf{f}(p_1), \quad p_1' = p_2' + \mathbf{b}(p_2'), \quad M_{\text{occ}} = [|p_1' - p_1| > t], \tag{4}$$

where $p_i$ denotes a pixel in image $I^i$, $\mathbf{f}$ and $\mathbf{b}$ represent forward and backward optical flows, respectively, and $t$ is an occlusion threshold.

**Dynamic Masks.** Dynamic regions, characterized by moving objects and non-rigid transformations, introduce significant challenges in accurately aligning pointmaps due to inconsistencies between camera-induced motion and object-specific motion. Without explicitly accounting for such dynamic behaviors, the network may attempt to incorrectly fit dynamic regions as if they were static, thus impairing reconstruction accuracy. To address this, we introduce a dynamic mask $M_{\text{dyn}}$ that explicitly highlights moving regions, guiding the network toward a more stable learning process. Specifically, the dynamic mask $M_{\text{dyn}}$ is computed by comparing optical flow $\mathbf{f}$ with the expected flow induced purely by camera motion $\mathbf{f}_{\text{cam}}$. Given the depth map $D$, intrinsics matrix $K$, relative rotation and translation $R, T$, and pixel coordinates $p$, we define:

$$\mathbf{f}_{\text{cam}} = \pi(DKRK^{-1}p + KT) - p, \quad M_{\text{dyn}} = [\|\mathbf{f}_{\text{cam}} - \mathbf{f}\| > \tau], \tag{5}$$

where $\pi : (x, y, z) \mapsto (x/z, y/z)$ is the projection operation, and $\tau$ serves as the dynamic threshold.

## 4.3 Objective Function

Given our SDAP representation equipped with occlusion and dynamic masks, we formulate our training objective as confidence-aware 3D regression [44], leveraging these masks to achieve enhanced stability. To effectively account for the distinct characteristics of static and dynamic regions, we introduce two separate objective functions, each explicitly designed to handle the respective scenarios and ensure accurate reconstruction in both cases.

**Pointmap alignment in static regions.** The regression loss in DUSt3R inherently aligns 3D pointmaps using camera pose alone. To ensure alignment focuses solely on static regions within image $I^2$, we employ the dynamic mask $M_{\text{dyn}}$ and restrict the computation of the regression loss

accordingly. Thus, we modify the regression loss $\mathcal{L}_{\text{regr}}$ as follows:

$$\mathcal{L}_{\text{regr}}(1,i) = \left\| \frac{1}{z} X_i^{1,1} - \frac{1}{\bar{z}} \bar{X}_i^{1,1} \right\|,$$

$$\mathcal{L}_{\text{regr}}(2,i) = \left(1 - M_{\text{dyn},i}^2\right) \left\| \frac{1}{z} X_i^{2,1} - \frac{1}{\bar{z}} \bar{X}_i^{2,1} \right\|. \tag{6}$$

To account for errorneously defined GT pointmaps, we additionally introduce a confidence-aware loss in static regions, $\mathcal{L}_{\text{static}}$, to incorporate uncertainties into the alignment process:

$$\mathcal{L}_{\text{static}} = \sum_{v \in \{1,2\}} \sum_{i \in \mathcal{D}^v} C_i^{v,1} \mathcal{L}_{\text{regr}}(v,i) - \alpha \log C_i^{v,1}. \tag{7}$$

**Pointmap alignment in dynamic regions.** To align dynamic region from $I^2$ to $I^1$, we introduce a dynamic alignment loss that leverages both the occlusion mask $M_{\text{occ}}$ and the dynamic mask $M_{\text{dyn}}$ to effectively address occlusions and motion. As illustrated in Figure 4, these masks are computed in a dedicated pipeline. This method ensures that when points from the second view $I^2$, are transformed into the coordinate system of $I^1$, they accurately correspond to the temporal state of the first view. In line with our regression loss, we further incorporate confidence estimates to define a confidence-aware alignment loss. The dynamic alignment loss is formulated as follows:

$$\mathcal{L}_{\text{dyn}} = \sum_{i \in \mathcal{D}^2} M_{\text{dyn},i}^2 (1 - M_{\text{occ},i}^2) C_i^{2,1} \left\| \frac{1}{\bar{z}_1} \bar{X}_{i+\mathbf{b}(i)}^{1,1} - \frac{1}{z_1} X_i^{2,1} \right\| - \alpha \log C_i^{2,1}$$

$$+ \sum_{i \in \mathcal{D}^1} M_{\text{dyn},i}^1 (1 - M_{\text{occ},i}^1) C_i^{1,2} \left\| \frac{1}{\bar{z}_2} \bar{X}_{i+\mathbf{f}(i)}^{2,2} - \frac{1}{z_2} X_i^{1,2} \right\| - \alpha \log C_i^{1,2}. \tag{8}$$

We leverage optical flow $\mathbf{f}$ to establish dense correspondences between $I^1$ and $I^2$, and its reverse direction $\mathbf{b}$. The first term in our loss function enforces the alignment of the pointmap in the coordinate space of the camera system associated with $I^1$, while the second term introduces a symmetric constraint by aligning the points when the roles of the views are swapped.

**Final Objective.** Our final objective function is defined as following:

$$\mathcal{L}_{\text{total}} = \mathcal{L}_{\text{static}} + \mathcal{L}_{\text{dyn}}. \tag{9}$$

This combined loss function enables our model to capture precise 3D geometry and robust correspondences in dynamic scenes while retaining DUSt3R's proven benefits in static regions.

## 4.4 Additional heads for downstream task

To further enhance the capabilities and interpretability of our SDAP framework, we introduce two additional downstream heads dedicated to explicitly modeling dynamic masks and optical flow.

**Dynamic mask head.** Since our model implicitly encodes regions corresponding to dynamic motion, we explicitly regress dynamic masks using an additional head. Specifically, we predict a single-channel logit map, $\hat{M}_{\text{dyn}}$, using a DPT head [32], structured similarly to our pointmap regression head. We supervise this head using a binary cross-entropy loss defined as follows:

$$\mathcal{L}_{\text{mask}} = -\frac{1}{|\mathcal{D}_{\text{all}}|} \sum_{i \in \mathcal{D}_{\text{all}}} \left[ M_{\text{dyn},i} \log(\sigma(\hat{M}_{\text{dyn},i})) + (1 - M_{\text{dyn},i}) \log(1 - \sigma(\hat{M}_{\text{dyn},i})) \right], \tag{10}$$

where $\sigma(\cdot)$ denotes the sigmoid function and $\mathcal{D}_{\text{all}}$ denotes the set of all pixels.

**Optical flow head.** To accurately estimate optical flow, we incorporate an additional head based on the RAFT architecture [40]. Inspired by recent findings from ZeroCo [1], our flow head utilizes cross-attention maps instead of traditional 4D correlation volumes. The optical flow head is supervised using the Mixture-of-Laplace loss [46].

# 5 Experiments

## 5.1 Experimental setup

**Implementation details.** Building on top of DUSt3R [44], we freeze the encoder and fine-tune only the decoder and the DPT head [32], as done similarly by MonST3R. For each epoch, we

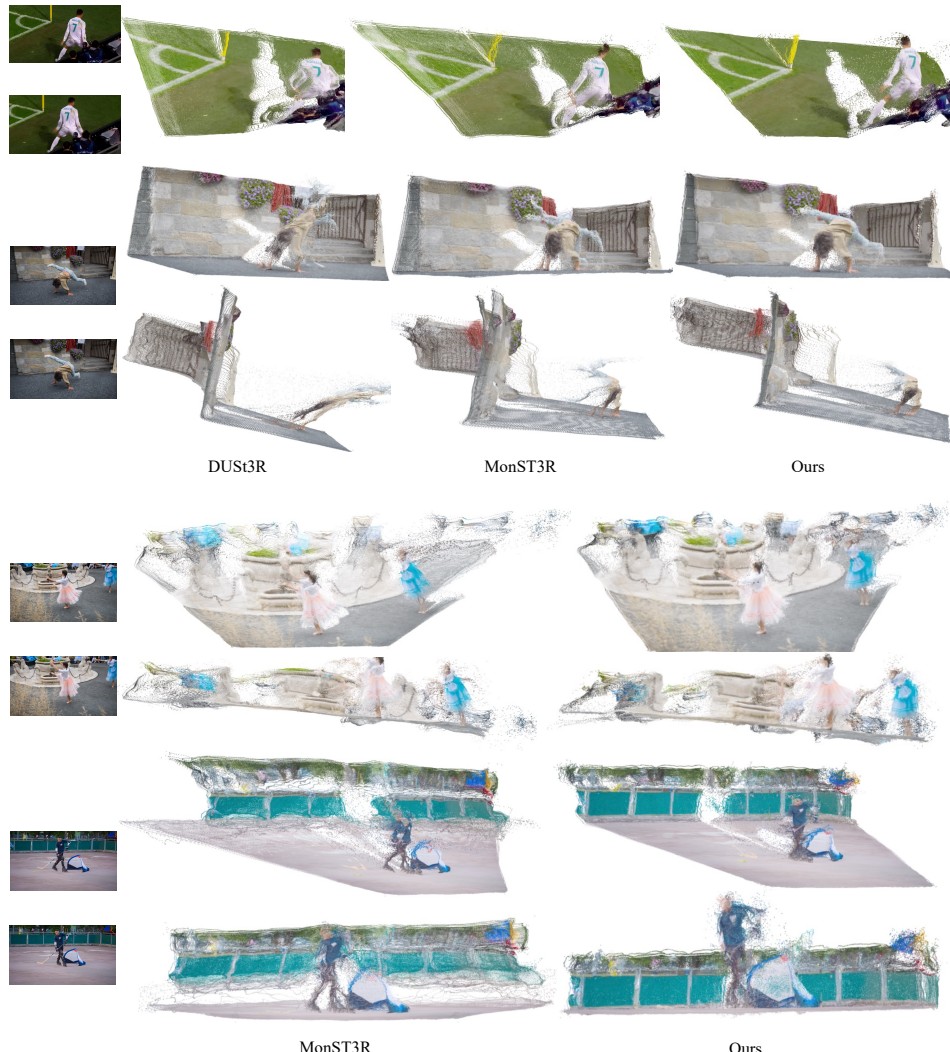

Figure 5: **Pointmap reconstruction.** We qualitatively compare the 3D pointmap of $\mathbf{D}^2\mathbf{USt3R}$ against other pointmap regression models [44, 51]. All visualizations presents per-pixel pointmaps without applying confidence thresholding. It is notable that both DUSt3R [44] and MonST3R [51] struggle to accurately reconstruct scenes that include dynamic object. We find that inaccurately established correspondence fields between dynamic regions negatively affect the overall reconstruction performance.

Table 2: **Multi-frame depth estimation results.** We compare multi-frame depth for both the entire scene and dynamic regions separately. The comparison for dynamic regions is conducted only when the dynamic parts are identifiable. *: Reproduced with same dataset as Ours.

| Category | Methods | TUM-Dynamics | | | | Bonn | | | | Sintel | | | | KITTI | |
| | | All | | Dynamic | | All | | Dynamic | | All | | Dynamic | | All | |
| | | AbsRel↓ | $\delta_1$↑ | AbsRel↓ | $\delta_1$↑ | AbsRel↓ | $\delta_1$↑ | AbsRel↓ | $\delta_1$↑ | AbsRel↓ | $\delta_1$↑ | AbsRel↓ | $\delta_1$↑ | AbsRel↓ | $\delta_1$↑ |
|---|---|---|---|---|---|---|---|---|---|---|---|---|---|---|---|
| Single-frame depth | DepthAnythingv2 [49] | 0.098 | 89.0 | - | - | 0.073 | 93.8 | - | - | 0.336 | 55.6 | - | - | 0.069 | 93.7 |
| | Marigold [19] | 0.205 | 72.3 | - | - | 0.066 | 96.4 | - | - | 0.623 | 50.5 | - | - | 0.104 | 89.9 |
| Multi-frame depth | DUSt3R [44] | 0.176 | 76.5 | 0.221 | 71.3 | 0.135 | 82.4 | 0.127 | 83.7 | 0.370 | **58.5** | 0.672 | **54.9** | 0.076 | 93.6 |
| | MASt3R [21] | 0.165 | 79.0 | 0.199 | 73.8 | 0.183 | 77.5 | 0.167 | 79.5 | 0.330 | 57.3 | 0.528 | 54.4 | **0.050** | **96.8** |
| | MonST3R [51] | 0.145 | 81.2 | 0.152 | 79.2 | 0.068 | 94.4 | 0.066 | 94.9 | 0.345 | 56.2 | **0.525** | 46.9 | 0.070 | 95.0 |
| | MonST3R* [51] | 0.159 | 81.0 | 0.181 | 76.5 | 0.076 | 93.9 | 0.071 | 94.4 | 0.349 | 52.5 | 0.565 | 36.9 | 0.103 | 90.9 |
| | **D$^2$USt3R (Ours)** | **0.142** | **83.9** | **0.148** | **82.9** | **0.060** | **95.8** | **0.059** | **95.7** | **0.324** | 57.5 | 0.568 | 48.0 | 0.104 | 90.7 |

randomly sample 20,000 image pairs and the network is trained for 50 epochs. We use the AdamW optimizer [27] with an initial learning rate of 5e-5. We train with 4 NVIDIA RTX 6000 GPUs, with a batch size of 4 images per GPU and gradient accumulation steps set to 2.

**Training datasets.** As shown in Table 1, we train $\mathbf{D}^2\mathbf{USt3R}$ on multiple datasets, including BlinkVision Outdoor [22], BlinkVision Indoor [22], Spring [29], PointOdyssey [54], and TartanAir [45]. Each epoch consisted of sampling 7,750, 4,750, 2,500, 2,500, and 2,500 pairs, respectively. Additionally,

Table 3: **Single-frame depth estimation results.** *: Reproduced using the same dataset as ours.

| Methods | Bonn | | Sintel | | KITTI | | NYU-v2 | | TUM-Dynamics | |
|---|---|---|---|---|---|---|---|---|---|---|
| | AbsRel↓ | $\delta_1$↑ | AbsRel↓ | $\delta_1$↑ | AbsRel↓ | $\delta_1$↑ | AbsRel↓ | $\delta_1$↑ | AbsRel↓ | $\delta_1$↑ |
| DUSt3R [44] | 0.141 | 82.5 | 0.424 | **58.7** | 0.112 | 86.3 | **0.080** | **90.7** | 0.176 | 76.8 |
| MASt3R [21] | 0.142 | 82.0 | 0.354 | 57.9 | **0.076** | **93.2** | 0.129 | 84.9 | 0.160 | 78.7 |
| MonST3R [51] | 0.076 | 93.9 | 0.345 | 56.5 | 0.101 | 89.3 | 0.091 | 88.8 | **0.147** | 81.1 |
| MonST3R* [51] | 0.083 | 93.6 | 0.387 | 50.6 | 0.143 | 85.0 | 0.084 | 90.1 | 0.163 | 79.1 |
| **D²USt3R (Ours)** | **0.065** | **95.2** | **0.340** | 58.4 | 0.131 | 86.2 | 0.085 | 90.1 | 0.150 | **82.9** |

Table 4: **Camera pose estimation results.** *: Reproduced using the same dataset as ours.

| Methods | Sintel | | | | TUM-Dynamics | | | | ScanNet | | | |
|---|---|---|---|---|---|---|---|---|---|---|---|---|
| | Rotation | | Translation | | Rotation | | Translation | | Rotation | | Translation | |
| | Avg↓ | Med↓ | Avg↓ | Med↓ | Avg↓ | Med↓ | Avg↓ | Med↓ | Avg↓ | Med↓ | Avg↓ | Med↓ |
| DUSt3R [44] | 6.15 | 4.51 | 0.29 | 0.26 | 2.36 | 0.98 | 0.013 | 0.01 | 0.74 | 0.54 | 0.11 | 0.08 |
| MASt3R [21] | 4.71 | 3.40 | 0.23 | 0.19 | 2.83 | 1.13 | 0.06 | 0.03 | 0.85 | 0.64 | 0.05 | 0.04 |
| MonST3R [51] | 4.90 | 2.30 | 0.26 | 0.22 | 1.88 | 1.39 | 0.019 | 0.01 | 0.94 | 0.79 | 0.10 | 0.08 |
| MonST3R* [51] | 8.50 | 2.61 | 0.27 | 0.23 | 1.76 | 1.40 | 0.02 | 0.01 | 0.74 | 0.58 | 0.10 | 0.08 |
| **D²USt3R (Ours)** | 6.96 | 2.67 | 0.26 | 0.22 | 1.80 | 1.41 | 0.03 | 0.02 | 0.75 | 0.57 | 0.08 | 0.06 |

Table 5: **Pointmap alignment accuracy in dynamic objects.** We report End-Point Error (EPE) ↓ on the Sintel and KITTI datasets. Note that Croco-Flow [47] is shown in gray on the Sintel benchmark because it was trained on that dataset. *: Reproduced using the same dataset as ours.

| Methods | Sintel-Clean | Sintel-Final | KITTI |
|---|---|---|---|
| DUSt3R [44] | 30.96 | 35.11 | 14.19 |
| MASt3R [21] | 39.37 | 39.50 | 13.27 |
| MonST3R [51] | 38.47 | 41.92 | 14.91 |
| MonST3R* [51] | 37.47 | 40.58 | 14.58 |
| **D²USt3R (Ours)** | 16.19 | 25.31 | 8.91 |
| Croco-Flow [47] | 3.31 | 4.28 | 13.24 |
| SEA-RAFT [46] | **5.21** | 13.18 | 4.43 |
| **D²USt3R + Flow head** | 9.25 | 12.77 | 3.57 |

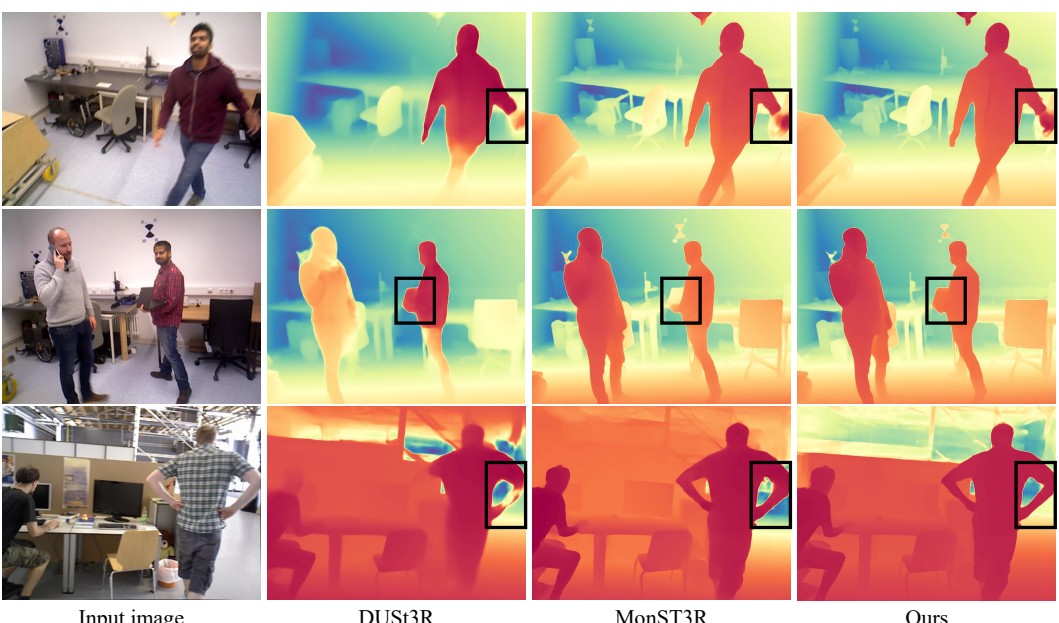

Figure 6: **Depth estimation qualitative results.**

Input image    DUSt3R    MonST3R    Ours

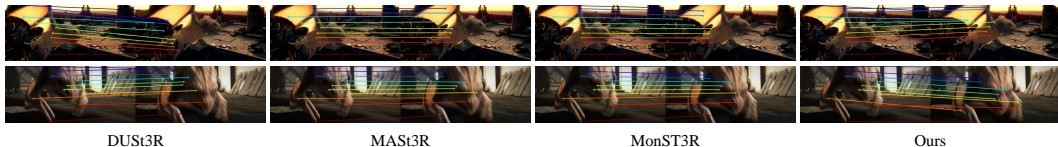

DUSt3R    MASt3R    MonST3R    Ours

Figure 7: **Visualization of correspondences given a pair of images.** We show that our method can find accurate correspondences between dynamic objects.

we perform random sampling with temporal strides varying from 1 to 9 to account for large camera motions and dynamic scenarios.

**Baselines.** Following [44], we evaluate our method on depth and camera pose estimation. We additionally evaluate pointmap alignment accuracy in dynamic regions. We compare our method against existing state-of-the-art pointmap regression models, specifically DUSt3R [44], MASt3R [21], and MonST3R [51]. Furthermore, to ensure fair comparisons and demonstrate the effectiveness of our approach, we trained a variant of MonST3R, termed MonST3R*, under the same setup as ours.

**Evaluation setup.** For multi-frame depth estimation, we evaluate on the TUM-Dynamics [38], Bonn [30], Sintel [2], KITTI [11], and ScanNet [7] datasets using image pairs with source frames offset from the target by strides of 1, 3, 5, 7, and 9 frames. We assess performance over the entire scene and exclusively on dynamic regions when dynamic masks are available. For single-

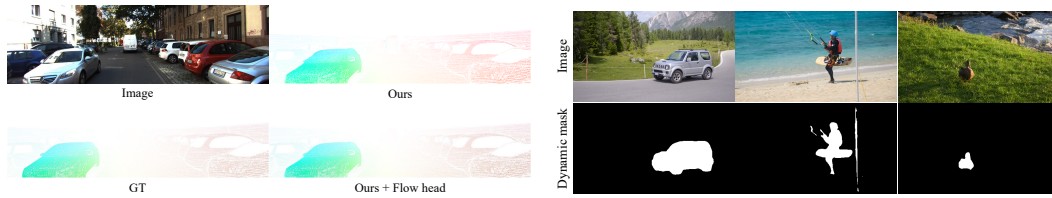

Figure 8: **Predicted optical flow.**    Figure 9: **Predicted dynamic mask.**

Table 6: **Robustness analysis on frame intervals.** We evaluate the robustness of our method with respect to varying frame intervals $\Delta t \in \{1, 3, 5, 7, 9\}$. *: Reproduced using the same dataset as ours.

| Methods | $\Delta t = 1$ | | $\Delta t = 3$ | | $\Delta t = 5$ | | $\Delta t = 7$ | | $\Delta t = 9$ | |
|---|---|---|---|---|---|---|---|---|---|---|
| | AbsRel↓ | $\delta_1$ ↑ | AbsRel↓ | $\delta_1$ ↑ | AbsRel↓ | $\delta_1$ ↑ | AbsRel↓ | $\delta_1$ ↑ | AbsRel↓ | $\delta_1$ ↑ |
| MonST3R*[51] | 0.078 | 93.8 | 0.078 | 93.5 | 0.075 | 93.9 | 0.074 | 93.9 | 0.072 | 94.3 |
| **$D^2USt3R$ (Ours)** | 0.061 | 95.6 | 0.061 | 95.5 | 0.060 | 95.7 | 0.061 | 95.9 | 0.058 | 96.2 |

frame depth estimation, we evaluate on the Bonn [30], Sintel [2], KITTI [10], NYU-v2 [36], and TUM-Dynamics [38] datasets. In both settings, we follow the affine-invariant depth evaluation protocol, reporting the Absolute Relative Error (AbsRel) and the percentage of inlier points ($\delta_1$, where $\delta < 1.25$).

## 5.2 Experimental results

**Depth estimation.** In this experiment, we evaluate our method and compare with existing methods on single/multi-frame depth estimation. The results are summarized in Table 2 and 3. We also show qualitative comparisons in Figure 5 and Figure 6. From the results, we observe that **$D^2USt3R$** outperforms other methods overall. However, our performance on KITTI is somewhat limited, likely due to the absence of driving scenes in our training data, which may have disadvantaged our model. Nonetheless, we compare with MonST3R*, which was trained with the same training datasets as ours, and find that the performance is comparable. Finally, we highlight in Figure 5 that our approach consistently predicts accurate depth for dynamic human subjects.

**Camera pose estimation.** In Table 4, we evaluate camera pose performance of our model on the Sintel [2], TUM-dynamics [38] and ScanNet [7] datasets. Note that we use the 2D–3D matching between the prediction $X^{2,1}$ and the original coordinates to obtain camera poses via PNP-RANSAC.

**Dynamic alignment.** In Table 5, we evaluate pointmap alignment accuracy in dynamic objects. For this, we use Sintel [2] and KITTI [10]. Since we directly obtain aligned pointmaps even in dynamic objects, we can easily derive 2D-2D matching points by computing the difference between $X^{2,1}$ and $X^{1,1}$. From the results, we find that our method outperforms other baselines, which is further supported in Figure 7, where accurately captured correspondences between objects in different time step frames are observed. This gap is further broadened when we leverage an additional flow head, as shown in Table 5, where ours with the flow head outperforms the state-of-the-art SEA-RAFT [46]. We provide qualitative examples in Figure 8.

**Dynamic mask.** In this experiment, we show that using dynamic mask head, our method reliably predicts dynamic regions. We show visualizations in Figure 9, where the dynamic mask head effectively segments dynamic objects across diverse in-the-wild scenarios.

## 5.3 Robustness analysis on frame intervals

We evaluate multi-frame depth estimation for the robustness of our method with respect to varying frame intervals on the Bonn dataset [30], as summarized in Table 6. Across all tested intervals ($\Delta t \in \{1, 3, 5, 7, 9\}$), the results show that our model maintains stable performance and even exhibits slight improvement as the interval widens, demonstrating strong temporal consistency and resilience to larger motion baselines. These results suggest that the proposed flow-chaining strategy effectively preserves geometric coherence across diverse motion dynamics, enabling reliable depth estimation under varying temporal conditions.

## 5.4 Ablation study

Although we've already validated our 3D alignment loss and effectiveness of proposed SDAP, we further examine how different fine-tuning strategies affect performance. Since our method, unlike DUSt3R or MonST3R, learns precise maps of cross-attention (Figure 3), we ask whether updating the encoder along with the decoder and head yields gains. As shown in Table 7, fine-tuning only the decoder and downstream head actually outperforms full fine-tuning. Therefore, in this work we restrict training to the decoder and head.

Table 7: **Ablation on training strategy.**

| Methods | TUM-Dynamics | | Bonn | |
|---|---|---|---|---|
| | AbsRel↓ | $\delta_1$ ↑ | AbsRel↓ | $\delta_1$ ↑ |
| Full finetune | 0.161 | 77.0 | 0.081 | 91.9 |
| Finetune decoder & head | 0.142 | 83.9 | 0.060 | 95.8 |

## 6 Conclusion

In this paper, we have introduced a novel approach for 3D dynamic scene reconstruction, featuring a simple yet effective extension to existing pointmap representations to accommodate dynamic motions. Our proposed method significantly enhances the quality of 3D reconstruction in dynamic environments. We evaluated our approach comprehensively across tasks including depth estimation, camera pose estimation, and 3D point alignment. Experimental results demonstrate that our method outperforms existing approaches on real-world, large-scale datasets, achieving new state-of-the-art performance.

## Acknowledgments and Disclosure of Funding

This research was supported by Institute of Information & communications Technology Planning & Evaluation (IITP) grant funded by the Korea government (MSIT) (RS-2019-II190075, RS-2024-00509279, RS-2025-II212068, RS-2023-00227592, RS-2025-02214479, RS-2024-00457882, RS-2025-25441838, RS-2025-25441838, RS-2025-02214479, RS-2025-02217259) and the Culture, Sports, and Tourism R&D Program through the Korea Creative Content Agency grant funded by the Ministry of Culture, Sports and Tourism (RS-2024-00345025, RS-2024-00333068, RS-2023-00222280, RS-2023-00266509), and National Research Foundation of Korea (RS-2024-00346597).

This research was supported by the ETH AI Center through an ETH AI Center postdoctoral fellowship to Sunghwan Hong.

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

# Enhancing 3D Reconstruction for Dynamic Scenes
## - Supplementary Materials -

**Jisang Han**[1*] **Honggyu An**[1*] **Jaewoo Jung**[1*] **Takuya Narihira**[2] **Junyoung Seo**[1]
**Kazumi Fukuda**[2] **Chaehyun Kim**[1] **Sunghwan Hong**[3] **Yuki Mitsufuji**[2,4†] **Seungryong Kim**[1†]

[1] KAIST AI  [2] Sony AI  [3] ETH Zürich AI Center, CVG, PRS  [4] Sony Group Corporation

https://cvlab-kaist.github.io/DDUSt3R

## A. Training dataset

Table 8: **List of excluded scenes from the datasets.**

| Dataset | Excluded Scenes |
|---|---|
| PointOdyssey | animal_s, animal_smoke_, animal1_s_, animal1_s, animal2_s, animal3_s, animal4_s, animal6_s, cab_e_ego2, cab_e1_3rd, cab_e1_ego2, cab_h_bench_3rd, cab_h_bench_ego2, character0_f, character0_f2, character3_f, character4_, character5_, character6, cnb_dlab_0215_3rd, cnb_dlab_0215_ego1, cnb_dlab_0225_3rd, cnb_dlab_0225_ego1, dancingroom_3rd, human_in_scene, kg, r5_new_f, scene_d78_0318_3rd, scene_d78_0318_ego1, scene_d78_0318_ego2, scene_j716_3rd, scene_j716_ego1, scene_j716_ego2, scene1_0129, seminar_h52_ego1 |
| Blinkvision Outdoor | outdoor_train_autopilot_tree_01, outdoor_train_autopilot_tree_02, outdoor_train_autopilot_tree_03, outdoor_train_track_animal_people |

We provide the scene names that were excluded during our training. These scenes are excluded, either because their annotations include errors. Please refer to Table 8.

## B. Bullet time reconstruction for dynamic video input

Owing to dynamic alignment and SDAP, we can aggregate and render a highly dynamic video input into a single, coherent bullet-time view. As illustrated in Figure 10, a video consisting of large input frames can be aligned into an intermediate bullet-time representation. This approach enables static reconstruction even when the input video includes dynamic motion. Consequently, it becomes feasible to directly train a 3D Gaussian splatting [20] on scenes containing moving people in landmarks or dynamics that are challenging to render using conventional methods.

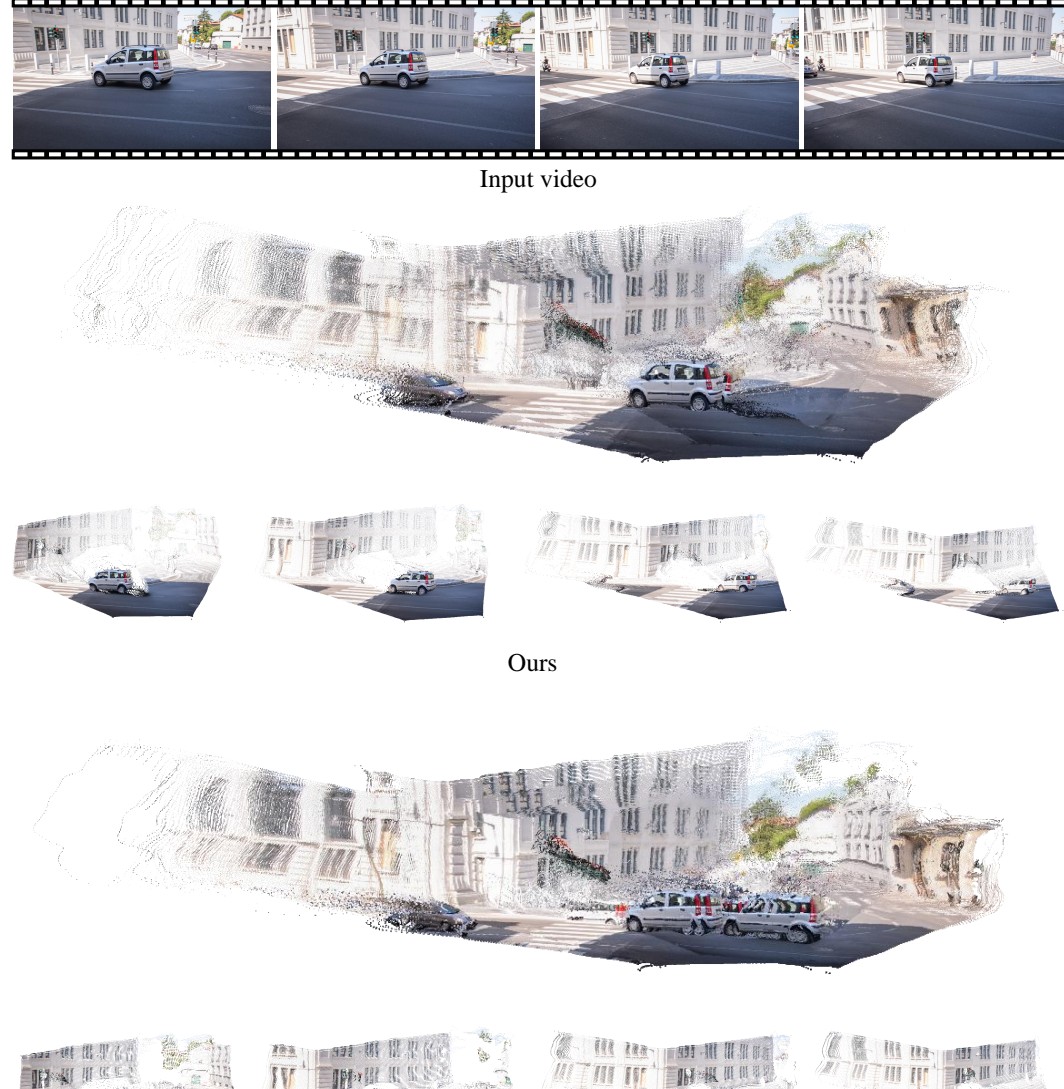

Input video

Ours

MonST3R

Figure 10: **Visualization of bullet-time reconstruction from a long sequence of dynamic video inputs.** We visualize the reconstruction of bullet-time view (20 frames) from a dynamic video input consisting of 40 frames. Since MonST3R is incapable of dynamic alignment, it predicts depth independently at each timestep, similar to monocular depth estimation, resulting in the reconstruction as a sequence of 3D pointmaps.

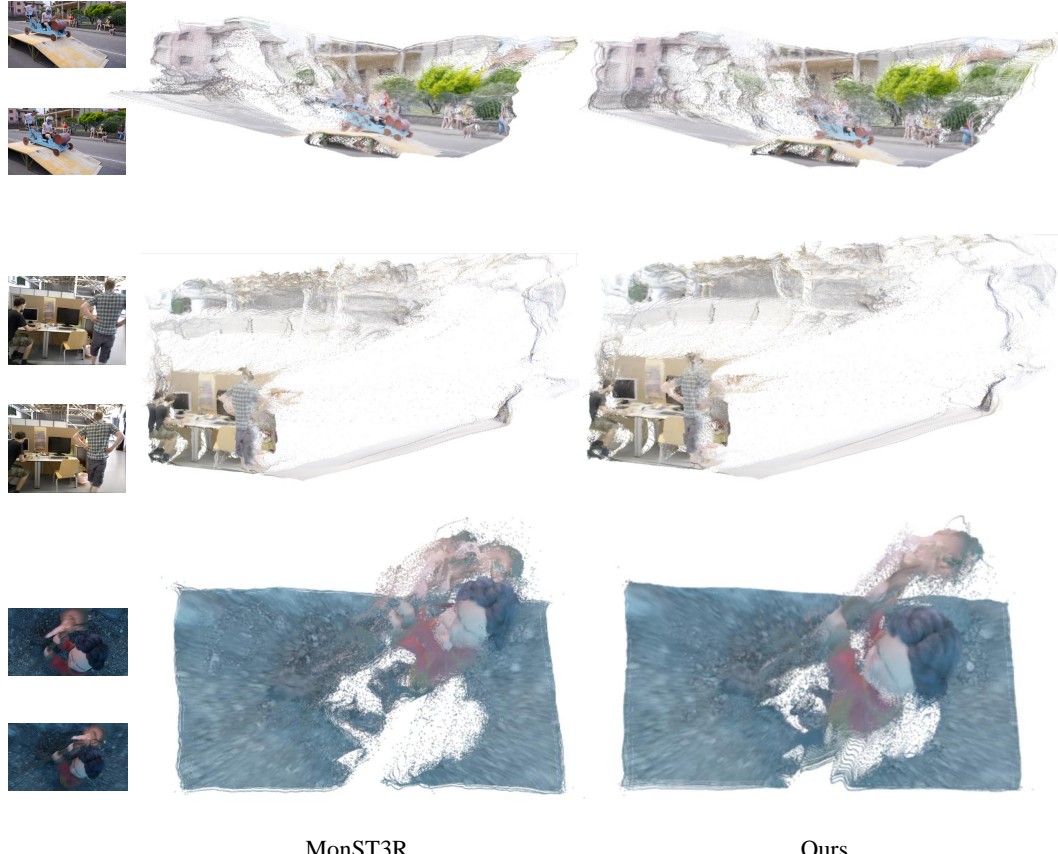

MonST3R                                        Ours

Figure 11: **Additional qualitative results for pointmap reconstruction.** We qualitatively compare the 3D pointmap of **D²USt3R** against MonST3R. All visualizations presents per-pixel pointmaps without applying confidence thresholding.

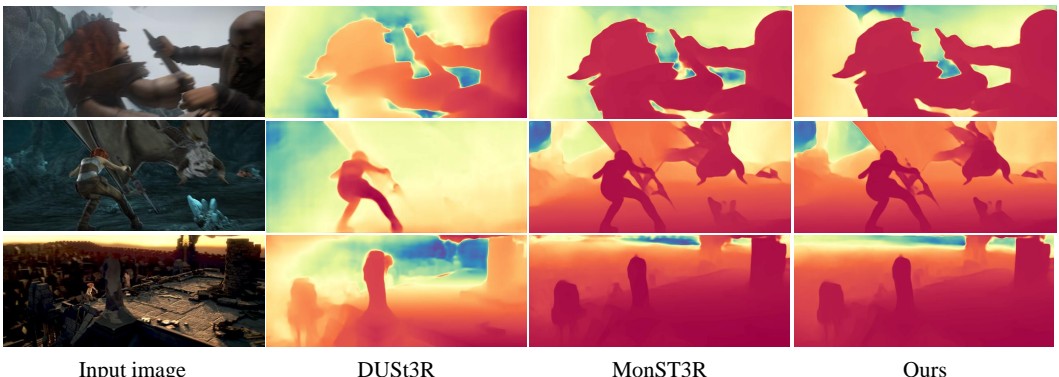

Input image              DUSt3R              MonST3R                Ours

Figure 12: **Additional qualitative results for depth estimation on Sintel dataset.**

## C. Additional results

In Figure 11 and Figure 12, we additionally present qualitative results for pointmap reconstruction and depth estimation. Utilizing our SDAP, dynamic elements are well-aligned, thereby enhancing both depth estimation and 3D reconstruction quality. In Figure 13 and Figure 14, we present qualitative results for optical flow on DAVIS [31] and KITTI [11] datasets. We observed that **D²USt3R** is capable of accurately predicting optical flow without relying on any dedicated optical flow module.

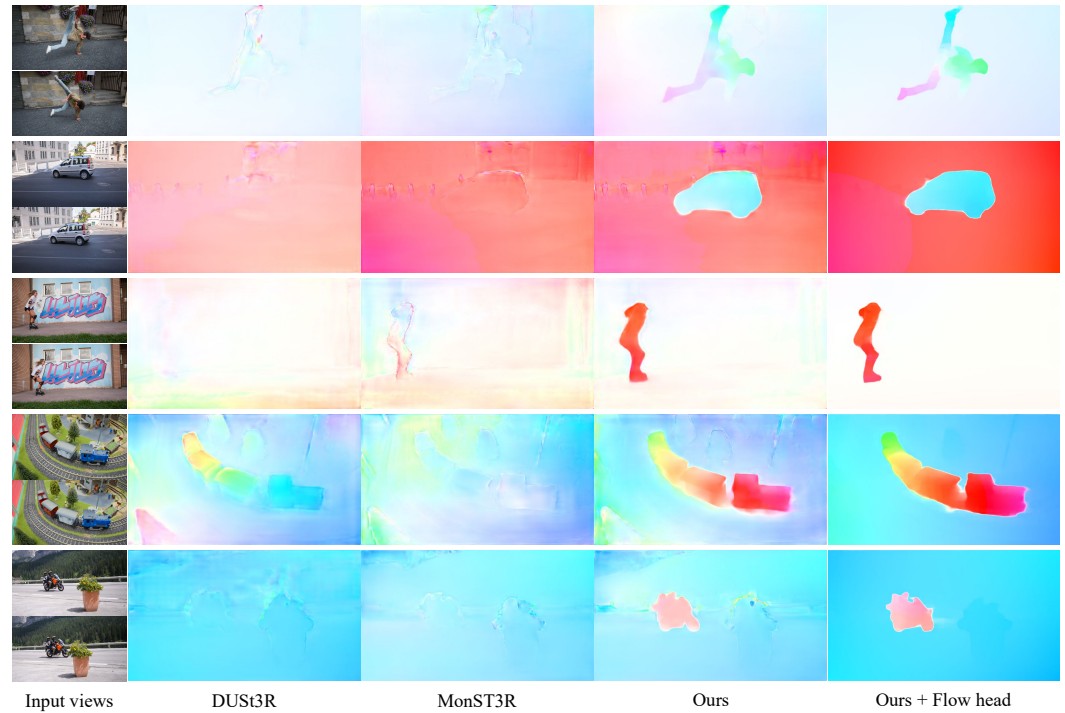

| Input views | DUSt3R | MonST3R | Ours | Ours + Flow head |

Figure 13: **Qualitative results for optical flow estimation on DAVIS dataset.**

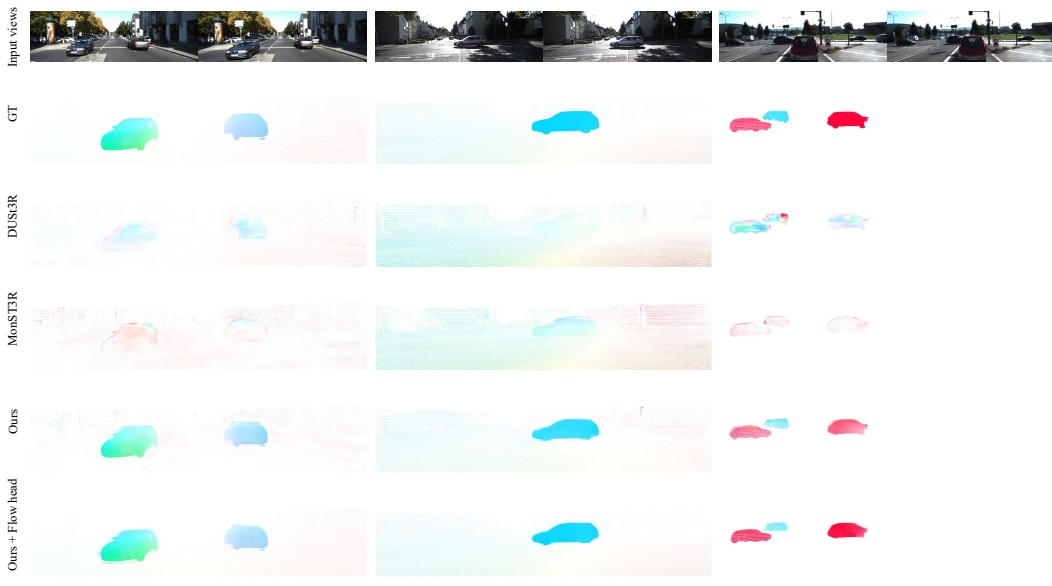

Figure 14: **Qualitative results for optical flow estimation on KITTI dataset.**

Table 9: **Multi-frame depth estimation results on end-to-end integration analysis.**

| Methods | TUM-Dynamics | | Bonn | | Sintel | | KITTI | |
|---|---|---|---|---|---|---|---|---|
| | AbsRel↓ | $\delta_1$ ↑ | AbsRel↓ | $\delta_1$ ↑ | AbsRel↓ | $\delta_1$ ↑ | AbsRel↓ | $\delta_1$ ↑ |
| **D$^2$USt3R** | 0.142 | 83.9 | 0.060 | 95.8 | 0.324 | 57.5 | 0.104 | 90.7 |
| **D$^2$USt3R** + Dynamic mask head | 0.140 | 84.4 | 0.059 | 95.6 | 0.313 | 58.3 | 0.105 | 90.3 |

## D. End-to-end dynamic mask head integration

We further explored an end-to-end training strategy in which the entire network, including the dynamic mask head, is jointly optimized. As shown in Table 9, this integrated setup encourages implicit interactions between sub-tasks through shared representations, leading to consistent, albeit moderate, performance improvements. Specifically, we observe gains on two of the four benchmarks, with a mean reduction in AbsRel of 1.7% and a mean increase in $\delta$ of 0.9 percentage points. These results confirm that joint optimization promotes beneficial cross-task interactions, yielding complementary improvements across depth and dynamic mask estimation.

## E. Limitations

Despite effectively handling dynamic scenarios, our dynamic alignment process has certain limitations. Since our model relies on an off-the-shelf optical flow model, its performance is dependent on the quality of the predicted optical flow. Additionally, our model is non-generative and can only take two frames as input. Because we aim to achieve alignment across time, extending our approach to video input would require an additional global optimization technique.

