# OpenReview forum: "Enhancing 3D Reconstruction for Dynamic Scenes"
_NeurIPS.cc/2025/Conference — NeurIPS 2025 poster_

### Official Review · Reviewer_JB4c · 2025-06-29

**Clarity:** 2
**Significance:** 2
**Originality:** 2
**Rating:** 4
**Confidence:** 4

**Summary:**

This paper introduces a framework for robust 3D reconstruction in dynamic scenes. It addresses the limitations of prior pointmap regression methods that struggle with object motion by proposing Static-Dynamic Aligned Pointmaps (SDAP). SDAP simultaneously accounts for both spatial structures and temporal motions, providing a more reliable 3D reconstruction of both static and dynamic regions. The method integrates occlusion and dynamic masks, derived from optical flow, to guide the network and stabilize learning. With a confidence-aware 3D regression loss that combines separate objectives for static and dynamic areas,this paper demonstrates superior 3D reconstruction performance across various complex motion datasets.

**Questions:**

Refer weaknesses

**Ethical Concerns:**

["NO or VERY MINOR ethics concerns only"]

**Final Justification:**

The new experiments with Segment Any Motion shows good results. Also the authors show additional comparison which seems to be convincing compared to earlier.

**Limitations:**

No. Integrate semantic segmentation models or object detection networks to provide high-level semantic and shape information, can make the dynamic mask generation more robust and accurate, especially for complex or ambiguous motions.

**Quality:**

2

**Strengths And Weaknesses:**

Strenth


1) Distinguishing camera-induced motion from genuine object-specific motion is challenging, as it requires precisely separating the camera’s ego-motion from the independent movements of objects in the scene.

2) It introduces a unique representation called Static-Dynamic Aligned Pointmaps, which can simultaneously account for both static spatial structures and dynamic temporal motions in a unified framework.

3) The paper is clearly written and easy to follow.



Weakness


1)The authors describe their dynamic mask generation as relying primarily on optical flow analysis, comparing the observed optical flow with that induced purely by camera motion. However, this approach does not inherently leverage high-level semantic or shape information. Modern state-of-the-art object tracking methods, such as SAMURAI, can effectively track dynamic objects in videos and could potentially address the challenges of dynamic mask generation; however, the authors did not conduct any study to evaluate this possibility.


2)The results in Table 2 show no improvement on the KITTI dataset, highlighting poor generalization. Despite KITTI’s mostly static scenes, the method still degrades, suggesting it fails to adapt when dynamic motion is absent.


3) There is a contradiction in the results: while Table 2 shows no improvement on the KITTI dataset, Table 5 reports a significant boost in pointmap alignment accuracy for dynamic objects. This suggests that the new method may actually degrade overall reconstruction quality.


4) The camera pose results in Table 4 do not show any improvement over the state of the art, and the authors fail to clearly explain this outcome.


5) The state-of-the-art methods considered are insufficient, as they do not include a comparison with 'Shape of Motion: 4D Reconstruction from a Single Video,' which performs reconstruction for long-range 3D tracking

---

> ### Author Rebuttal · Authors · 2025-07-31
>
> > [W1, L1] The authors describe their dynamic mask generation as relying primarily on optical flow analysis, comparing the observed optical flow with that induced purely by camera motion. However, this approach does not inherently leverage high-level semantic or shape information. Modern state-of-the-art object tracking methods, such as SAMURAI, can effectively track dynamic objects in videos and could potentially address the challenges of dynamic mask generation; however, the authors did not conduct any study to evaluate this possibility.
> >
>
> We thank the reviewer for the valuable feedback that can further strengthen our method. Below, we adopt Segment Any Motion in Videos [1] that leverages DINOv2 and 2D tracks for dynamic mask prediction, and use it to replace the dynamic masks we acquired from our approach. Below summarizes the results.
>
> | Methods (AbsRel / $\delta$) | TUM  | Bonn | Sintel | KITTI |
> | --- | --- | --- | --- | --- |
> | Ours | 0.142 / 83.9 | 0.060 / 95.8 | 0.324 / 57.5 | 0.104 / 90.7 |
> | Ours + Segment Any Motion | 0.140 / 84.4 | 0.062 / 95.4 | 0.313 / 58.4 | 0.104 / 90.5 |
>
> From the experiment, we confirm that our method can benefit from more advanced dynamic mask methods, leaving a room for further improvement. We will include this discussion in the final version.
>
> [1] Huang, Nan, et al. "Segment Any Motion in Videos." CVPR'25.
>
> > [W2] The results in Table 2 show no improvement on the KITTI dataset, highlighting poor generalization. Despite KITTI’s mostly static scenes, the method still degrades, suggesting it fails to adapt when dynamic motion is absent.
> >
>
> We would like to clarify that **it is natural and reasonable to expect and observe similar to Monst3R for KITTI dataset, as the reviewer said, it is mostly static, where SDAP is identical to Monst3R’s objective function**. This is denoted as Monst3R*, which shows apparent difference to original Monst3R that was specifically trained with the driving scene dataset. its domain is significantly different from that of the other datasets. As mentioned in Line 208, our model was not trained on any driving scenes, so it is difficult to expect performance improvements on this dataset.
>
> > [W3] There is a contradiction in the results: while Table 2 shows no improvement on the KITTI dataset, Table 5 reports a significant boost in pointmap alignment accuracy for dynamic objects. This suggests that the new method may actually degrade overall reconstruction quality.
> >
>
> We apologize for the confusion and appreciate the opportunity to clarify the apparent discrepancy. **Table 2 and Table 5 evaluate two different tasks under distinct protocols.** Table 2 follows the standard multi‑frame **depth‑estimation benchmark**, which scores per‑pixel depth on the reference frames. Because ground‑truth depths in this benchmark are provided almost exclusively for static, first‑surface points, every method—ours included—is evaluated mainly on regions where object motion is minimal. Consequently, the numerical differences between our approach and strong baselines are necessarily small.
>
> **Table 5, by contrast, measures pointmap alignment error across time, explicitly including the dynamic regions detected by our masks.** Baseline methods do not enforce correspondences on moving objects, so their alignment accuracy drops sharply in these areas. Our motion‑aware objective, however, keeps the pointmaps coherent even when objects move, leading to the much larger performance gap reported in Table 5. The two tables are therefore complementary: Table 2 reflects conventional static‑depth accuracy, while Table 5 highlights our method’s key advantage—robust temporal alignment in scenes with motions.
>
> > [W4] The camera pose results in Table 4 do not show any improvement over the state of the art, and the authors fail to clearly explain this outcome.
> >
>
> We appreciate the reviewer’s concern about camera-pose accuracy. In 3D reconstruction in dynamic scenes, pose estimates obtained with a single PnP-RANSAC pass are often sub-optimal for systems which handle correspondences in dynamic regions. To mitigate this issue, recent systems such as **Shape-of-Motion**, **Spatial Tracker v2**, and **MegaSAM** tackle this by iteratively refining the pose by masking out dynamic regions at each step, or feed-forward frameworks like Dust3R and Monst3R do not address correspondences in dynamic regions in their 3D pointmap representation.
>
> Our goal, however, is to keep the pipeline **fully feed-forward and lightweight while also modeling correspondences in dynamic regions**. We introduce a unified representation (SDAP) that predicts geometry for both static and dynamic content while embedding camera intrinsics and extrinsics in a single pass. To preserve this simplicity we use the **vanilla PnP-RANSAC**, foregoing specialized pose modules or iterative motion segmentation.
>
> Although this design sacrifices some of the accuracy gains that iterative schemes can deliver, it yields a streamlined, scalable model that still matches—or surpasses—state-of-the-art results on several benchmarks. We will add this trade-off discussion to the final paper and clarify that incorporating auxiliary heads for flow, tracking, or explicit pose refinement is an exciting direction for future work. We will include this discussion in the final version.
>
> > [W5] The state-of-the-art methods considered are insufficient, as they do not include a comparison with 'Shape of Motion: 4D Reconstruction from a Single Video,' which performs reconstruction for long-range 3D tracking
> >
>
> We would like to emphasize that our primary goal is to demonstrate that our SDAP brings improvements in 3D reconstruction for dynamic scenes when trained on the same dataset. While *Shape of Motion* performs reconstruction and 2D/3D tracking from video, it relies on scene-specific optimization methods and primarily focuses on tracking and novel view synthesis, which is quite different from our approach. As a more reasonable comparison, we additionally include evaluations of multi-frame depth, camera pose, and pointmap alignment accuracy against Align3r[1] and MegaSAM[2], which are more aligned with our method. The results are presented below.
>
> [Multi-frame depth estimation]
>
> | Methods (AbsRel / $\delta$) | Bonn | Bonn (dynamic) | Sintel | Sintel (dynamic) |
> | --- | --- | --- | --- | --- |
> | Align3r | 0.067 / 95.4 | 0.075 / 94.5 | 0.353 / 59.8 | 0.543 / 48.1 |
> | MegaSAM | 0.079 / 93.6 | 0.086 / 91.6 | 0.424 / 65.0 | 0.635 / 57.5 |
> | Ours | 0.060 / 95.8 | 0.059 / 95.7 | 0.324 / 57.5 | 0.568 / 48.0 |
>
> [Camera pose estimation]
>
> | Methods (mean / median) | Sintel (rotation) | Sintel (Translation) |
> | --- | --- | --- |
> | Align3r | 8.50 / 2.61 | 0.27 / 0.23 |
> | MegaSAM | 6.67 / 2.46 | 0.32 / 0.23 |
> | Ours | 6.96 / 2.67 | 0.26 / 0.22 |
>
> Additionally, in contrary to our novel SDAP representation that simultaneously capture both static and dynamic 3D scene geometry, Align3R models pointmaps generated by a single global rigid transformation, leading to compromised correspondence learning, which is supported by the table below.
>
> [Pointmap alignment accuracy]
>
> | Methods (EPE) | Sintel (clean) | Sintel (final) |
> | --- | --- | --- |
> | Align3r | 38.64 | 39.89 |
> | Ours | 16.19 | 25.31 |
>
> Moreover, we additionally include speed and efficiency comparisons for completeness.
>
> [Time complexity and Memories]
>
> | Methods | Time (s) | Memories (MB) |
> | --- | --- | --- |
> | Align3r | 2.166 | 10447 |
> | MegaSAM | 13.33 | 7364 |
> | Ours | 0.226 | 3483 |
>
> To summarize, our novel approach to adopt SDAP enables simultaneous modeling of both static and dynamic 3D geometry, showing competitive performance in multi-frame depth and pose estimation while significantly improving the performance in dynamic region alignment, with improved efficiency and speed.
>
> [1] Lu, Jiahao, et al. "Align3r: Aligned monocular depth estimation for dynamic videos." CVPR'25
>
> [2] Li, Zhengqi, et al. "MegaSaM: Accurate, fast and robust structure and motion from casual dynamic videos." CVPR’25

---

> ### Comment · Reviewer_JB4c · 2025-08-05
> **After rebuttle**
>
> Dear Authors,
> Thanks for your rebuttal. This addresses many of my concerns. Raising the score.

---

> > ### Author Response · Authors · 2025-08-06
> >
> > Dear Reviewer JB4c,
> >
> > Thank you for your thoughtful review and for taking the time to engage with our work. We truly appreciate your response and feedback.

---

### Official Review · Reviewer_kbwQ · 2025-07-01

**Clarity:** 2
**Significance:** 3
**Originality:** 3
**Rating:** 4
**Confidence:** 4

**Summary:**

This paper trains a neural network to estimate a point map for each frame from two given video frames that capture dynamic objects. The network learns how to estimate depth for different types of motion. To this end, the static background and the dynamic object regions are separated and are trained with sophisticated objectives that address occlusion and dynamic regions. The qualitative and quantitative comparisons demonstrate its performance across various dynamic scene datasets.

**Questions:**

I have the following questions:

- Please clarify the notation of a predicted point map X^{n,m} and explain whether there is a conflict between the two objective terms mentioned above.

- In the supplemental, the reconstruction in the background suffers from wavy artifacts and a multiview inconsistency problem. Does the proposed method show degraded performance for the static dataset? Figure 5 also shows that the Dust3r achieves reasonable relative scene scales and a field of view on the static region comparable to Monst3r and the proposed method. It would be great to conduct the evaluation for the static region and the dynamic region individually.

- Do the predicted optical flow and depth agree with each other?

- Does the proposed method produce temporal consistency and multiview consistency? Since the video is not produced, those components cannot be evaluated.

**Ethical Concerns:**

["NO or VERY MINOR ethics concerns only"]

**Final Justification:**

The reviewers addressed my concerns with thorough experiments. This makes me lean toward accepting this paper. If the authors had submitted the supplemental video in advance, the reviewers could have appreciated the performance of this paper at a glance. In the final version, I highly recommend including the supplemental video of the proposed method.

**Limitations:**

Yes.

**Paper Formatting Concerns:**

None.

**Quality:**

2

**Strengths And Weaknesses:**

The paper’s strengths are as follows;

- The performance outperforms other learning-based stereo models, such as Dust3r and Monst3r, for monocular and binocular depth estimation.
- The occlusion mask and dynamic mask are utilized to train the network without false guidance. For a dynamic region, the objective encourages the network to estimate a point map at the time as its counterpart, shown in Equation 8.
- Supplemental shows the outperformance of the proposed method against Monst3r.

There are the following weak points:

- Term X^{1,2} in Equation 8 is used, but there is no explicit definition of it. This makes me confused about understanding the training objectives. Specifically, the first term in Equation 6 and the second term in Equation 8 could conflict with each other if X_1,2 and X_1,1 are from the same estimation.
- Term X has different meanings at Line 89 and Line 92. It should be consistent throughout the paper context.
- The optical flow supervision is available for synthetic datasets. This may lead to a domain gap when the trained model is applied to a real-world dataset.
- The proposed model doesn’t utilize the predicted optical flow for depth estimation. Plus, optical flow and depth estimation are computed independently. This may cause the inconsistency between two different types of predictions: optical flow and 3d points. However, it has not been discussed in the paper.
- The evaluation doesn't separate the dynamic and the static regions.
- Figure 7 is too small to see the correspondence matches.
- There is no video demo that can verify the temporal consistency of the proposed method.
- The comparison for camera pose estimation is not plausibly discussed.

---

> ### Author Rebuttal · Authors · 2025-07-31
>
> > [W1, Q1] No definition of X^{1,2}. Specifically, the first term in Equation 6 and the second term in Equation 8 could conflict with each other if X_1,2 and X_1,1 are from the same estimation. Please clarify the notation of a predicted point map X^{n,m} and explain whether there is a conflict between the two objective terms mentioned above.
> >
>
> Our definition of the pointmap strictly follows the notation used in DUSt3R. As described in Line 89, X^{n,m} denotes the 3D point map predicted from image I^n, but expressed in the camera coordinate frame of I^m. Therefore, X^{1,2} denotes the 3D pointmap predicted from I^1, expressed in the camera coordinate frame of I^2.
>
> Regarding the potential conflict raised by the reviewer, the term X_{1,2} in Equation 8 refers to the pointmap obtained by swapping the input image pair during inference which is from different estimation, as explained in Line 163. Therefore, X_{1,2} and X_{1,1} in the first term of Equation 6 are not the same estimation in different coordinate frames, but are instead distinct predictions resulting from different input orderings. Additionally, in Equation 8, X_{1,2} refers to the pointmap predicted from I^1, where the dynamic regions are estimated to be aligned from the temporal state of I^2. We apologize for any confusion caused by the lack of precise clarification in our notation and we will revise the manuscript accordingly to improve clarity.
>
> > [W2] Term X has different meanings at Line 89 and Line 92.
> >
>
> We apologize for the confusion. The term X at Line 89 should be corrected to $\bar{X}$. We will revise the manuscript accordingly.
>
> > [W3] The optical flow supervision is available for synthetic datasets. This may lead to a domain gap when the trained model is applied to a real-world dataset.
> >
>
> As dynamic datasets with ground-truth depth and camera pose are difficult to obtain, we used only synthetic datasets for training. This has been **a common practice in numerous works** [1,2,3,4,5], as real-world data captured with sensors is typically noisy and sparse, which may be detrimental to network learning rather than helping. Prior works [1,2,3,4,5] also show that this convention is effective in generalizing well to real-world datasets. This is supported by our evaluation in Tab. 2,3,4,5, in which TUM-Dynamics and Bonn contain challenging factors such as camera/motion blur, yet our model was still able to generalize well under these conditions, further validating its robustness.
>
> [1] Li, Zhengqi, et al. "MegaSaM: Accurate, fast and robust structure and motion from casual dynamic videos." CVPR’25
>
> [2] Lu, Jiahao, et al. "Align3r: Aligned monocular depth estimation for dynamic videos." CVPR'25
>
> [3] Wang, Qianqian, et al. "Tracking everything everywhere all at once." ICCV’23.
>
> [4] Wang, Qianqian, et al. "Shape of motion: 4d reconstruction from a single video." arXiv’24
>
> [5] Zhang, Junyi, et al. "Monst3r: A simple approach for estimating geometry in the presence of motion." ICLR’25
>
> > [W4, Q3] The proposed model doesn’t utilize the predicted optical flow for depth estimation. Plus, optical flow and depth estimation are computed independently. This may cause the inconsistency between two different types of predictions: optical flow and 3d points. However, it has not been discussed in the paper. Do the predicted optical flow and depth agree with each other?
> >
>
> We wish to highlight that our novel SDAP representation aims to encapsulates depth, camera intrinsic, extrinsic and flow. The separate optical flow head is designed for modeling better smoothness prior, but this still stems from our shared feature representation that also outputs the SDAP. However, we agree that there may exist inconsistency between our SDAP output and the optical flow from the separate head. Nevertheless, **this is also shared by the recent works CoPoNeRF [1] and VGGT [2]**, the radiance field does not align perfectly with the flows and where pointmap does not perfectly align with depth and camera pose outputs. Still, the predictions are within reasonable range, making predictions do not exhibit significant inconsistencies.  We will also include this discussion in the final version.
>
> Additionally, acting on the reviewer’s suggestion, we could try to train the entire network—including the dynamic mask head ,in a single end-to-end manner, to encourage implicit interactions between the predictions via shared representation. This setup forces the intermediate feature tensor to act as a shared representation for the different heads, enabling complementary supervisory signals to interact implicitly.  The corresponding results are summarized in the table below.
>
> | Methods (AbsRel / $\delta$) | TUM-Dynamics | Bonn | Sintel | KITTI |
> | --- | --- | --- | --- | --- |
> | Ours | 0.142 / 83.9 | 0.060 / 95.8 | 0.324 / 57.5 | 0.104 / 90.7 |
> | Ours + integrated | 0.140 / 84.4 | 0.059 / 95.6 | 0.313 / 58.3 | 0.105 / 90.3 |
>
> Even though only the dynamic mask head was jointly optimized, we still observed performance improvements—end-to-end training delivers consistent, if moderate, gains on two of the four benchmarks (mean AbsRel ↓ 1.7 %, mean δ ↑ 0.9 pp), confirming that joint optimization promotes the tasks to improve one another—an effect also reported in [1, 2,3].
>
> However, we regret that due to limited time and resources available for this rebuttal, we were unable to include the optical flow head in this end-to-end training. However, building upon a well established empirical knowledge that correspondence-based supervisory signals for multi-task learning formulations can enhance more reliable 3D reconstruction [1, 2], we expect there would exist potential enhancements if we include additional heads, such as tracking, flows and camera pose estimation.
>
> [1] Hong, Sunghwan, et al. "Unifying correspondence, pose and nerf for pose-free novel view synthesis from stereo pairs." CVPR'24.
>
> [2] Wang, Jianyuan, et al. "Vggt: Visual geometry grounded transformer." CVPR’25.
>
> > [W5, Q2] The evaluation doesn't separate the dynamic and the static regions.
> >
>
> We wish to clarify that in **Tab. 2, we have separated the evaluation metric to All and Dynamic, which should already reflect the reviewer’s comment**. Nevertheless, we conduct an additional experiment, where below shows **evaluations solely performed on static regions**.
>
> | Methods (AbsRel / $\delta$) | TUM-Dynamics | Bonn | Sintel | KITTI |
> | --- | --- | --- | --- | --- |
> | Monst3r | 0.147 / 83.2 | 0.061 / 95.8 | 0.313 / 56.4 | N/A |
> | Ours | 0.137 / 85.3 | 0.059 / 96.1 | 0.323 / 55.3 | N/A |
>
> From the results, we find that the results are comparable to Monst3R on static regions, adhering to our SDAP formulation where the static regions are supervised identically to the original DUSt3R. By also supervising dynamic regions so that both static and dynamic regions are aligned to their corresponding counterparts, we achieve similar performance in static regions while largely improving the reconstruction quality in dynamic regions.
>
> > [W6] Fig7 is too small.
> >
>
> We apologize for the inconvenience. We will revise it accordingly.
>
> > [W7, Q4] No video demo that can verify the temporal consistency of the proposed method. Does the proposed method produce temporal consistency and multiview consistency? Since the video is not produced, those components cannot be evaluated.
> >
>
> We would like to clarify that, as noted in the **supplementary material**, the current method operates on frame pairs because our primary contribution is the **Static‑Dynamic Aligned Pointmap (SDAP),** a novel representation that jointly models static geometry and dynamic motion. Extending SDAP to true multi‑view or sliding‑window optimization for temporal consistency is an orthogonal direction that we are actively investigating and view as promising future work.
>
> > [W8] Comparison for camera pose estimation is not plausibly discussed.
> >
>
> We appreciate the reviewer’s concern about camera-pose accuracy. In 3D reconstruction in dynamic scenes, pose estimates obtained with a single PnP-RANSAC pass are often sub-optimal for systems which handle correspondences in dynamic regions. To mitigate this issue, recent systems such as **Shape-of-Motion**, **Spatial Tracker v2**, and **MegaSAM** tackle this by iteratively refining the pose by masking out dynamic regions at each step, or feed-forward frameworks like Dust3R and Monst3R do not address correspondences in dynamic regions in their 3D pointmap representation.
>
> Our goal, however, is to keep the pipeline **fully feed-forward and lightweight while also modeling correspondences in dynamic regions**. We introduce a unified representation (SDAP) that predicts geometry for both static and dynamic content while embedding camera intrinsics and extrinsics in a single pass. To preserve this simplicity we use the **vanilla PnP-RANSAC**, foregoing specialized pose modules or iterative motion segmentation.
>
> Although this design sacrifices some of the accuracy gains that iterative schemes can deliver, it yields a streamlined, scalable model that still matches—or surpasses—state-of-the-art results on several benchmarks. We will add this trade-off discussion to the final paper and clarify that incorporating auxiliary heads for flow, tracking, or explicit pose refinement is an exciting direction for future work. We will include this discussion in the final version.

---

> > ### Comment · Reviewer_kbwQ · 2025-08-05
> > **Comments**
> >
> > Thank the authors for the rebuttal. Here I attach additional comments.
> >
> > [W4, Q3] The proposed model doesn’t utilize the predicted optical flow for depth estimation. Plus, optical flow and depth estimation are computed independently. This may cause the inconsistency between two different types of predictions: optical flow and 3d points. However, it has not been discussed in the paper. Do the predicted optical flow and depth agree with each other?
> >
> > Comments: Thank the authors for the additional experiments. However, as this paper addresses dynamic scenes, exploring optical flow seems mandatory.
> >
> > [W5, Q2] The evaluation doesn't separate the dynamic and the static regions.
> >
> > Comment: Figure 5 shows the clear difference between methods for the background. In the top of Figure 5, it seems Duster correctly estimates the background while Monster and the proposed method suffer from the perspective error. What makes this difference?  And, it would be great to include Duster's quantitative results.
> >
> > [W7, Q4] No video demo that can verify the temporal consistency of the proposed method. Does the proposed method produce temporal consistency and multiview consistency? Since the video is not produced, those components cannot be evaluated.
> >
> > Comment: At least, evaluation should be demonstrated in this paper as a limitation.
> >
> > [W8] Comparison for camera pose estimation is not plausibly discussed.
> >
> > We appreciate the reviewer’s concern about camera-pose accuracy. In 3D reconstruction in dynamic scenes, pose estimates obtained with a single PnP-RANSAC pass are often sub-optimal for systems which handle correspondences in dynamic regions. ...
> >
> > Comment: The experiment above and the camera pose estimation experiment in the paper have some inconsistencies. For example, Monster has less accuracy for TUM-Dynamics and Sintel for static estimation. About pose estimation, Monster achieves better pose estimation for Sintel and TUM-dynamics (translations). I think the common belief is that the performance of pose estimation relies on the accuracy of the static region, but the data doesn't support this. What makes this difference?

---

> > > ### Author Response · Authors · 2025-08-06
> > > **Response (1/3)**
> > >
> > > > Thank the authors for the additional experiments. However, as this paper addresses dynamic scenes, exploring optical flow seems mandatory.
> > >
> > > We emphasize that optical-flow  is already a core element of our SDAP framework. As demonstrated in Eq. 8, Fig. 8, and Tab. 5, we incorporate flow both in the loss function and through a dedicated prediction head—unlike Monst3R, which leaves optical flow entirely unexplored. Although limited rebuttal time prevented us from jointly training the flow head, prior work shows that correspondence-based supervision consistently strengthens 3D reconstruction [1, 2]. To compensate for the lacking joint optimization for flow head, we conducted an additional experiment with a jointly trained dynamic-mask head, which yielded moderate gains. We therefore expect that extending joint training to the flow head would further improve performance as it has been often observed that multi-task learning helps, as well as [1,2] showed such supervisions help in 3D reconstruction task. We wish to assure that we plan to include these results in the final version.
> > >
> > > Furthermore, during this rebuttal period, we have also explored the robustness of our method to noisy flows:
> > >
> > > To deepen the analysis of our method’s robustness to optical‑flow noise, we had conducted a controlled study in which we inject uniformly sampled noise within [-1, 1] range into the flow fields, systematically degrading their quality. For each noise level we retrain and evaluate the model, enabling a direct, quantitative comparison across flow qualities. The “SEA-RAFT” row represents the common case where ground‑truth flow is unavailable, while the last row represents when the same model is driven by noisy flows, emulating the use of less capable off‑the‑shelf estimators.
> > >
> > > | Methods (AbsRel / $\delta$) | TUM-Dynamics | Bonn | Sintel | KITTI |
> > > | --- | --- | --- | --- | --- |
> > > | SEA-RAFT | 0.142 / 83.9 | 0.060 / 95.8 | 0.324 / 57.5 | 0.104 / 90.7 |
> > > | SEA-RAFT+ 1 px noise | 0.143 / 83.8 | 0.063 / 95.5 | 0.346 / 52.8 | 0.106 / 90.5 |
> > >
> > > | Methods (mean / medium) | Sintel (rotation) | Sintel (Translation) | Tum (rotation) | Tum (translation) |
> > > | --- | --- | --- | --- | --- |
> > > | SEA-RAFT | 6.96 / 2.67 | 0.26 / 0.22 | 1.80 / 1.41 | 0.03 / 0.02 |
> > > | SEA-RAFT+ 1 px noise | 6.51 / 3.09 | 0.27 / 0.23 | 1.89 / 1.44 | 0.03 / 0.02 |
> > >
> > > From the results, we find that adding noises leads to comparable or only slightly degraded performance. Despite injecting noise levels twice as high as those observed in SEA-RAFT predictions, the model still learns reasonably robust representations.
> > >
> > > Below, we conduct an additional robustness check – accuracy of SEA‑RAFT flow versus ground‑truth (GT).
> > >
> > > | EPE | BlinkVision Outdoor | BlinkVision Indoor | TartanAir | Sintel |
> > > | --- | --- | --- | --- | --- |
> > > | Ours vs GT Flow (EPE) | 1.44 | 1.14 | 1.18 | 0.74 |
> > >
> > > Across the four diverse benchmarks the end‑point error (EPE) stays tightly clustered around a single‑pixel discrepancy (mean ≈ 1.13 px). On 288×512 frames this is < 0.2 % of the image diagonal, implying that the off‑the‑shelf SEA‑RAFT flow is already close to the information content of GT flow, and adopting cycle consistency further reduces the error, concluding that  substituting GT with SEA‑RAFT is unlikely to introduce a training bias attributable to flow quality, corroborating the earlier noise‑injection study.
> > >
> > > Given that (i) we already evaluate optical flow, (ii) incorporate it directly in the loss, (iii) introduce a dedicated optical-flow head, and (iv) conducted additional experiments presented above, we believe **our work has explored optical-flow supervision thoroughly and convincingly.**

---

> > > ### Author Response · Authors · 2025-08-06
> > > **Response (2/3)**
> > >
> > > > Figure 5 shows the clear difference between methods for the background. In the top of Figure 5, it seems Duster correctly estimates the background while Monster and the proposed method suffer from the perspective error. What makes this difference? And, it would be great to include Duster's quantitative results.
> > >
> > > Regarding the qualitative example in Figure 5, both MonST3R and our method were **fine-tuned with much smaller datasets compared to DUSt3R and also our dataset comprises of dynamic scenes**. This can naturally lead to differences from the original DUSt3R, especially once dynamic content is introduced—where factors like occlusions and clutters caused by motion can affect performance, and this is natural as it is generally hard to assume perfect data that covers above challenges. Nonetheless, our primary focus is on aligning dynamic regions, and we have demonstrated that our approach outperforms MonST3R when trained on the same dataset. Incorporating static background consistency into our framework could further strengthen our method, and we will consider this direction in future work.
> > >
> > > Still, as suggested by the reviewer, we have included DUSt3R in the static region evaluation, and the results are shown below.
> > > | Methods (AbsRel / $\delta$) | TUM-Dynamics | Bonn | Sintel | KITTI |
> > > | --- | --- | --- | --- | --- |
> > > | Dust3r | 0.155 / 79.1 | 0.088 / 90.1 | 0.291 / 57.3 | N/A |
> > > | Monst3r | 0.147 / 83.2 | 0.061 / 95.8 | 0.313 / 56.4 | N/A |
> > > | Ours | 0.137 / 85.3 | 0.059 / 96.1 | 0.323 / 55.3 | N/A |
> > >
> > > From the results, we draw the same conclusion: we find that the results are comparable to both Dust3r and Monst3r on static regions, adhering to our SDAP formulation where the static regions are supervised identically to the original DUSt3R. By also supervising dynamic regions so that both static and dynamic regions are aligned to their corresponding counterparts, we achieve similar performance in static regions while largely improving the reconstruction quality in dynamic regions.
> > >
> > > > At least, evaluation should be demonstrated in this paper as a limitation.
> > >
> > > Thank you for the helpful suggestion. As we noted in both the paper and the rebuttal, our present contribution centres on **Static–Dynamic Aligned Pointmaps (SDAP)**—a representation designed to align static and dynamic regions within individual view pairs.
> > >
> > > While a global N-view optimization (e.g., the strategy adopted in Monst3R) could in principle be grafted onto SDAP for the evaluation that the reviewer suggests, doing so would require **non-trivial changes to our pipeline and might blur the paper’s main message: how SDAP improves dynamic-region alignment**. For the sake of conceptual clarity, we therefore chose to defer a full multi-view extension to future work. We do, however, recognise its importance and plan to explore joint alignment across multiple views in a follow-up study, where its impact can be isolated and assessed thoroughly.

---

> ### Author Response · Authors · 2025-08-06
> **Response (3/3)**
>
> > The experiment above and the camera pose estimation experiment in the paper have some inconsistencies. For example, Monster has less accuracy for TUM-Dynamics and Sintel for static estimation. About pose estimation, Monster achieves better pose estimation for Sintel and TUM-dynamics (translations). I think the common belief is that the performance of pose estimation relies on the accuracy of the static region, but the data doesn't support this. What makes this difference?
>
> Thank you for raising this discrepancy.
>
> As mentioned by the reviewer, we agree that the key factor of camera pose accuracy is **the reliability of correspondences in the *static* parts of the scene**, which directly affects robust solvers such as PnP + RANSAC.  When *all* observed points are static, this rule of thumb generally holds: better static separation → cleaner correspondences → better pose.  However, in mixed scenes (e.g., Sintel, TUM-Dynamics) the solver also receives spurious matches from dynamic objects, occlusions, and parallax effects.  These outliers degrade pose accuracy even if the static region separation itself is good, leading to the seemingly contradictory numbers you highlighted.  This effect is consistent with the robust-estimation principles discussed in standard multi-view-geometry references [1] and has been empirically analyzed in dynamic-SLAM literature.
>
> Because of this brittleness, a growing body of work—e.g., “shape-of-motion”, “MegaSAM”, and the dynamic-SLAM literature—incorporates *iterative* or *joint* optimization to down-weight dynamic correspondences.  In contrast, purely feed-forward pipelines like ours and Monst3R rely on a single pass of classical robust solvers, and the community is still exploring how to make these solvers resilient to dynamic clutter.  We therefore consider closing this gap an interesting avenue for future work. In addition, although both ours and Monst3r are purely feed-forward frameworks, Monst3r does not estimate any dynamic correspondences, which leads to much less distractors when estimating camera pose using PnP + RANSAC. For example, for a scene of a man waving his right hand, Monst3r will simply estimate two hands in different 3D positions, whereas our estimation will align these hands by also considering dynamic correspondences. Therefore, Monst3r does not need to consider any spurious dynamic correspondences, as they can only obtain correspondences from static regions.
>
> Finally, we would like to emphasize that our main contribution is **Static–Dynamic Aligned Pointmaps (SDAP)**.  Unlike static-only pointmaps, SDAP aligns both static and dynamic regions, encoding each pixel’s space-time information in a unified representation.  As SDAP still depends on standard pose solvers, it is reasonable that our raw pose numbers are similar to those of Monst3R; the novelty lies in how SDAP *uses* those poses for richer 3D reasoning, not in proposing a new pose-estimation algorithm itself.
>
> [1] R. Hartley and A. Zisserman, *Multiple View Geometry in Computer Vision*, 2nd ed., Cambridge UP, 2003.

---

> > ### Comment · Reviewer_kbwQ · 2025-08-06
> >
> > Thank the author for the quick answers.
> >
> > I have one more question. Compared to https://easi3r.github.io/, what is the benefit of the proposed method?

---

> ### Author Response · Authors · 2025-08-06
>
> We thank the reviewer for another question.
>
> EASI3R, **a concurrent work of ours that was realeased on arxiv after March 1st**, aims to reconstruct dynamic scene by masking out dynamic regions with proposed approach that leverages inherent information within attention maps, extending DUSt3R to dynamic scenes in a training-free manner. In contrast, our approach introduces a novel formulation—Static–Dynamic Aligned Pointmap (SDAP)—which explicitly aligns both **static and dynamic regions** to enable coherent 3D reconstruction across the entire scene.
>
> While EASI3R attempts to efficiently mask out the dynamic regions by leveraging emergent properties within attention maps, it does not distinguish itself from Dust3R's approach that fundamentally relies only on correspondences from static regions. By masking out dynamic regions before pointmap estimation, the reconstruction for those masked areas must rely solely on monocular depth priors, limiting accuracy and spatial consistency, especially in highly dynamic scenarios.In contrast, our SDAP framework explicitly accounts for correspondences within both static and dynamic regions. This allows the model to jointly reason about scene geometry and object motion, resulting in more accurate and robust 3D reconstructions. **SDAP representation is, again, the key contribution and difference that makes our approach unique from others.**
>
> Thanks to such newly proposed formulations, once the SDAP is estimated, our method can directly provide dense correspondences within dynamic regions as part of the output, something that other works did not account for. Rather than discarding motion information, our method incorporates it explicitly, enabling more robust modeling of challenging scenes with multiple moving objects.
>
> We believe this principled difference highlights the strength of our approach in handling dynamic scenes with more comprehensive and learned geometric understanding.

---

> > ### Comment · Reviewer_kbwQ · 2025-08-07
> >
> > Thank you.
> >
> > By the way, the proposed method relies on the accuracy of the optical flow estimator. I am wondering how much displacement could be tolerated by the proposed method. As I know, the optical flow estimator works well only for small camera or object motions. As the teaser suggests that this paper targets big displacements, reporting the accuracy regarding the time gap between two frames would be helpful.

---

> ### Author Response · Authors · 2025-08-07
>
> Thank you for the additional question. We agree with the reviewer that optical-flow estimators become less reliable as the baseline between frames increases. To mitigate this error propagation during training, we adopt a **flow-chaining** strategy: rather than estimating flow directly across distant frames, we compose a series of short-baseline flows, which are markedly more accurate. We will include this in the implementation detail.
>
> We also ran a dedicated robustness study in which we injected controlled noise into the chained flows; the downstream network exhibited only a modest performance drop (see Response 1/3 in our previous rebuttal), confirming its tolerance for typical flow imperfections.
>
> As requested, we now break down multi-frame depth accuracy by frame interval Δt on the Bonn dataset. The headline figure reported in the main paper is simply the mean of these entries.
>
> [Multi-frame depth estimation on the Bonn dataset (Absrel/ $\delta$)]
>
> |  Δt | 1 | 3 | 5 | 7 | 9 |
> | --- | --- | --- | --- | --- | --- |
> | Monst3r | 0.078 / 0.938 | 0.078 / 0.935 | 0.075 / 0.939 | 0.074 / 0.939 | 0.072 / 0.943 |
> | Ours | 0.061 / 0.956 | 0.061 / 0.955 | 0.060 / 0.957 | 0.061 / 0.959 | 0.058 0.962 |
>
> Unexpectedly, **performance improves as the interval widens**. Empirically, we have noticed that Dust3R also performs sub-optimally on extremely small baselines but steadily improves as the baseline widens—up to a point beyond which correspondences break down. The trend in above table mirrors this observation. We will include this discussion in the supplementary material for completeness.

---

> > ### Author Response · Authors · 2025-08-08
> >
> > Thank you again for your valuable and insightful feedback. We would greatly appreciate it if you could let us know whether your concerns have been addressed. If you have any additional questions, we will be happy to address them.

---

### Official Review · Reviewer_tcmj · 2025-07-02

**Clarity:** 4
**Significance:** 3
**Originality:** 3
**Rating:** 4
**Confidence:** 4

**Summary:**

The paper proposes $D^2UST3R$, an approach to improve upon previous 3D aware pointmap regression approaches such as Dust3r and Monst3r for dynamic scenes. They propose SDAP: Static-Dynamic Aligned Pointmaps, which capture both static and dynamics elements of the 3d geometry. They also propose leveraging an alignment loss that takes into account missing direct 3d correspondences due to occlusion or motion. Finally, they benchmark across a variety of tasks such as camera pose estimation, multi-frame depth, etc.

**Questions:**

1. I would like to see comparisons to some more recent methods for dynamic scene reconstruction as stated in the weaknesses.
2. How does inaccurate flow impact depth or pose estimation, for eg if you were to use a weaker model than the current one your approach is using?
3. Have you tested on real dynamic-scene datasets with hand-held cameras such as DAVIS or Waymo? How does the model perform in those cases?

**Ethical Concerns:**

["NO or VERY MINOR ethics concerns only"]

**Final Justification:**

I have read through the responses and thank the authors for providing a detailed rebuttal. The responses seemed to have given more clarity about the positioning and experiments of the paper. Therefore, I'm holding my original score.

**Limitations:**

1. Please address the point about not using real-world dataset in training. Is that by design or some other reason?

**Quality:**

3

**Strengths And Weaknesses:**

Strengths
1. The paper is written well and easy to follow
2. Unified modelling of motion and static elements(SDAP) is an interesting approach to handle dynamic scenes


Weaknesses
1. While the method improves upon 3D aware regression, it still seems imperative to compare to methods such as Align3R[1], MegaSAM[2], etc.
2. Quality of Flow: How robust is the approach towards noisy flow? Currently the method relies on ground truth flow and off-the-shelf state of the art flow estimators.
3. All five training datasets are synthetic. No real-capture dynamic-scene sequence seems to be used for training so generalizability to real photometrics, motion-blur, sensor noise is unproven.

[1]Lu, Jiahao, et al. "Align3r: Aligned monocular depth estimation for dynamic videos." Proceedings of the Computer Vision and Pattern Recognition Conference. 2025.

[2] Li, Zhengqi, et al. "MegaSaM: Accurate, fast and robust structure and motion from casual dynamic videos." Proceedings of the Computer Vision and Pattern Recognition Conference. 2025.

---

> ### Author Rebuttal · Authors · 2025-07-31
>
> > [W1, Q1] While the method improves upon 3D aware regression, it still seems imperative to compare to methods such as Align3R[1], MegaSAM[2], etc. I would like to see comparisons to some more recent methods for dynamic scene reconstruction as stated in the weaknesses.
> >
>
> We thank the reviewer for the constructive suggestion. Due to the long runtime required for MegaSAM’s test-time optimization, we provide comparison results on the Bonn and Sintel datasets. Also we provide pointmap alignment accuracy on the Sintel dataset. The results are shown below:
>
> [Multi-frame depth estimation]
>
> | Methods (AbsRel / $\delta$) | Bonn | Bonn (dynamic) | Sintel | Sintel (dynamic) |
> | --- | --- | --- | --- | --- |
> | Align3r | 0.067 / 95.4 | 0.075 / 94.5 | 0.353 / 59.8 | 0.543 / 48.1 |
> | MegaSAM | 0.079 / 93.6 | 0.086 / 91.6 | 0.424 / 65.0 | 0.635 / 57.5 |
> | Ours | 0.060 / 95.8 | 0.059 / 95.7 | 0.324 / 57.5 | 0.568 / 48.0 |
>
> [Camera pose estimation]
>
> | Methods (mean / median) | Sintel (rotation) | Sintel (Translation) |
> | --- | --- | --- |
> | Align3r | 8.50 / 2.61 | 0.27 / 0.23 |
> | MegaSAM | 6.67 / 2.46 | 0.32 / 0.23 |
> | Ours | 6.96 / 2.67 | 0.26 / 0.22 |
>
> Additionally, in contrary to our novel SDAP representation that simultaneously capture both static and dynamic 3D scene geometry, Align3R models pointmaps generated by a single global rigid transformation, leading to compromised correspondence learning, which is supported by the table below.
>
> [Pointmap alignment accuracy]
>
> | Methods (EPE) | Sintel (clean) | Sintel (final) |
> | --- | --- | --- |
> | Align3r | 38.64 | 39.89 |
> | Ours | 16.19 | 25.31 |
>
> Moreover, we additionally include speed and efficiency comparisons for completeness.
>
> [Time complexity and Memories]
>
> | Methods | Time (s) | Memories (MB) |
> | --- | --- | --- |
> | Align3r | 2.166 | 10447 |
> | MegaSAM | 13.33 | 7364 |
> | Ours | 0.226 | 3483 |
>
> To summarize, our novel approach to adopt SDAP enables simultaneous modeling of both static and dynamic 3D geometry, showing competitive performance in multi-frame depth and pose estimation while significantly improving the performance in dynamic region alignment, with improved efficiency and speed.
>
> > [W2, Q2] Quality of Flow: How robust is the approach towards noisy flow? Currently the method relies on ground truth flow and off-the-shelf state of the art flow estimators. How does inaccurate flow impact depth or pose estimation, for eg if you were to use a weaker model than the current one your approach is using?
> >
>
> Thank you for the constructive feedback.
>
> We acknowledge that supervision derived from off‑the‑shelf optical flow models could, in principle, limit performance (However, please note that leveraging off-the-shelf flow models has been a common practice in many prior works, e.g., [1, 2]). In our setting, however, this risk is minimized for three reasons:
>
> 1. **Quality of optical flow.** Following previous works, our work mostly utilizes **synthetic data** to train with accurate depth, optical flow, and camera pose information, which is typically hard to accurately obtain in real-world videos. As a result, only 12.5% of our training batch utilizes flow from off-the-shelf models (Point Odyssey dataset), where the remaining flow is obtained from ground-truth information.
> 2. **Uncertainty‑aware point‑map regression.** In addition to the majority of the data coming from ground-truth optical flow, in our framework, the flow provides soft correspondences that are blended with multiview cues; erroneous matches and potentially noisy flows from off-the-shelf models are down‑weighted through the uncertainty term in our loss, as in DUSt3R.
> 3. **Empirical evidence.** Our experiments show that the uncertainty‑aware point‑map regression effectively suppresses errors near depth discontinuities and occlusions by assigning low confidence to those regions.
>
> We have prepared visualizations that highlight these low‑confidence predictions along boundaries and occlusions through our empirical studies. Although the rebuttal format does not allow external media, we will include these visuals in the final version of the paper.
>
> To deepen the analysis of our method’s robustness to optical‑flow noise, we conduct a controlled study in which we inject uniformly sampled noise within [-1, 1] range into the flow fields, systematically degrading their quality. For each noise level we retrain and evaluate the model, enabling a direct, quantitative comparison across flow qualities. The row (“SEA-RAFT”) corresponds to the default setting that relies on a standard off‑the‑shelf optical‑flow estimator—representing the common case where ground‑truth flow is unavailable. The last row shows performance when the same model is driven by noisy flows, emulating the use of weaker off‑the‑shelf estimators. Below summarizes the resulting metrics.
>
> | Methods (AbsRel / $\delta$) | TUM-Dynamics | Bonn | Sintel | KITTI |
> | --- | --- | --- | --- | --- |
> | SEA-RAFT | 0.142 / 83.9 | 0.060 / 95.8 | 0.324 / 57.5 | 0.104 / 90.7 |
> | SEA-RAFT+ 1 px noise | 0.143 / 83.8 | 0.063 / 95.5 | 0.346 / 52.8 | 0.106 / 90.5 |
>
> | Methods (mean / medium) | Sintel (rotation) | Sintel (Translation) | Tum (rotation) | Tum (translation) |
> | --- | --- | --- | --- | --- |
> | SEA-RAFT | 6.96 / 2.67 | 0.26 / 0.22 | 1.80 / 1.41 | 0.03 / 0.02 |
> | SEA-RAFT+ 1 px noise | 6.51 / 3.09 | 0.27 / 0.23 | 1.89 / 1.44 | 0.03 / 0.02 |
>
> From the results, we find that adding noises leads to comparable or only slightly degraded performance. Despite injecting noise levels twice as high as those observed in Sea-RAFT predictions (Sea-RAFT noise + additional noise; the error of SEA-RAFT is roughly 1px compared to ground-truth shown in the table below), the model still learns reasonably robust representations. Consequently, our empirical evidence and comprehensive robustness evaluations strongly support that our framework effectively minimizes the impact of optical-flow inaccuracies, even when utilizing less accurate, off-the-shelf estimators.
>
>
>
> Below, we conduct an additional robustness check – accuracy of Sea‑RAFT flow versus ground‑truth (GT).
>
> | EPE | BlinkVision Outdoor | BlinkVision Indoor | TartanAir | Sintel |
> | --- | --- | --- | --- | --- |
> | Ours vs GT Flow (EPE) | 1.44 | 1.14 | 1.18 | 0.74 |
>
> Across the four diverse benchmarks the end‑point error (EPE) stays tightly clustered around a single‑pixel discrepancy (mean ≈ 1.13 px). On 288×512 frames this is < 0.2 % of the image diagonal, implying that the off‑the‑shelf Sea‑RAFT flow is already close to the information content of GT flow, and adopting cycle consistency further reduces the error. We believe that the advances in the optical-flow field will further refine our pseudo-ground truth optical flow, narrowing the already small gap to perfect accuracy.
>
> > [W3, L1] All five training datasets are synthetic. No real-capture dynamic-scene sequence seems to be used for training so generalizability to real photometrics, motion-blur, sensor noise is unproven. Please address the point about not using real-world dataset in training. Is that by design or some other reason?
> >
>
> As dynamic datasets with ground-truth depth and camera pose are difficult to obtain, we used only synthetic datasets for training. This has been a **common practice in numerous works** [1,2,3,4,5], as real-world data captured with sensors is typically noisy and sparse, which may be detrimental to network learning rather than helping. Prior works [1,2,3,4,5] also show that this convention is effective in generalizing well to real-world datasets. This is supported by our evaluation in Tab. 2,3,4,5, in which TUM-Dynamics and Bonn contain challenging factors such as camera/motion blur, yet our model was still able to generalize well under these conditions, further validating its robustness.
>
> [1] Li, Zhengqi, et al. "MegaSaM: Accurate, fast and robust structure and motion from casual dynamic videos." CVPR’25
>
> [2] Lu, Jiahao, et al. "Align3r: Aligned monocular depth estimation for dynamic videos." CVPR'25
>
> [3] Wang, Qianqian, et al. "Tracking everything everywhere all at once." ICCV’23.
>
> [4] Wang, Qianqian, et al. "Shape of motion: 4d reconstruction from a single video." arXiv’24
>
> [5] Zhang, Junyi, et al. "Monst3r: A simple approach for estimating geometry in the presence of motion." ICLR’25
>
> > [Q3] Have you tested on real dynamic-scene datasets with hand-held cameras such as DAVIS or Waymo? How does the model perform in those cases?
> >
>
> We would like to clarify that the TUM-Dynamics and Bonn datasets used in our evaluation are real-world dynamic scene datasets. For the DAVIS dataset, quantitative evaluation is not feasible due to the lack of ground-truth depth. However, as shown in the qualitative results in Figure 5, which includes the pointmap reconstructions from a YouTube real-world video and DAVIS scenes, our method demonstrates strong performance even in real dynamic scenes.

---

### Official Review · Reviewer_YZSu · 2025-07-03

**Clarity:** 3
**Significance:** 3
**Originality:** 3
**Rating:** 4
**Confidence:** 3

**Summary:**

The paper proposes an enhanced 3D reconstruction method for dynamic scenes, addressing limitations in existing approaches like DUS3R and MonST3R, which struggle with dynamic object motion due to their reliance on static scene assumptions. The authors introduce Static-Dynamic Aligned Pointmaps (SDAP), a novel representation that integrates camera pose information for static regions and optical flow for dynamic regions to achieve consistent 3D point correspondences across frames. By incorporating motion-aware training objectives, the method improves depth estimation and pointmap alignment in dynamic scenes. The approach is evaluated on datasets like TUM-Dynamics, Sintel, and KITTI, demonstrating superior performance in multi-frame depth estimation and pointmap alignment compared to baselines. Training details, including datasets and hyperparameters, are provided, though pretrained weights are not released due to their size.

**Questions:**

Please see the weaknesses.

**Ethical Concerns:**

["NO or VERY MINOR ethics concerns only"]

**Final Justification:**

My concerns have been addressed, maintaining the score.

**Limitations:**

yes

**Quality:**

3

**Strengths And Weaknesses:**

Strengths
- The introduction of Static-Dynamic Aligned Pointmaps is a significant contribution, effectively addressing the misalignment issues in dynamic scenes by combining camera pose-based alignment for static regions with optical flow-based warping for dynamic regions. This dual approach is a clever solution to the temporal inconsistency problem.
- The paper conducts thorough evaluations across multiple datasets (TUM-Dynamics, Sintel, KITTI) with varying temporal strides, providing robust evidence of the method’s effectiveness in both static and dynamic regions. The inclusion of qualitative visualizations (Figures 5-7) strengthens the interpretability of results.
- By augmenting the feed-forward framework with explicit motion-aware constraints, the method tackles the limitations of per-frame training paradigms in prior work, leading to improved consistency in depth estimation and pointmap reconstruction.
- The paper provides sufficient details on dataset construction, loss functions, and training procedures (Sections 4.2, 4.3, 4.4, 5.1), facilitating potential reproducibility despite the lack of code release.

Weaknesses
- The paper provides an "end-to-end" solution for dynamic scene reconstruction. However, the core component—Static-Dynamic Aligned Pointmap (SDAP)—relies heavily on external optical flow estimators (RAFT or precomputed ground-truth flow). This design introduces additional computational overhead and potential inconsistencies, and ultimately constrains the upper bound of model performance due to its dependence on the quality of the external flow. Moreover, the paper does not provide an in-depth analysis of the robustness of the reconstruction to flow errors, nor does it evaluate system performance under scenarios with missing or inaccurate optical flow.
- The authors introduce an occlusion mask to filter out unalignable regions. However, the results lack any quantitative evaluation of reconstruction quality specifically within occluded areas. Given that occlusion is often the most challenging aspect in dynamic scene reconstruction, this omission limits the assessment of the method’s robustness and generalization to boundary conditions.
- The authors design separate losses for static and dynamic regions (L_static and L_dyn), yet the alignment loss for dynamic regions depends on warping via optical flow indices. In practice, this approach is prone to sparsity and misalignment issues, especially in occluded regions, at depth discontinuities, or near object boundaries. The resulting loss becomes unstable and may introduce noise into the overall reconstruction. Furthermore, this split-loss design fails to consider continuity and smoothness across static-dynamic region boundaries, which can lead to “crack-like” artifacts in the reconstruction.
- The authors introduce additional heads—Dynamic Mask Head and Optical Flow Head—aiming to support dynamic region recognition and auxiliary supervision. However, these modules are trained independently and are not integrated into the SDAP optimization process in a closed-loop manner. This lack of interaction limits their effectiveness and wastes the potential of mutual reinforcement.
- The loss function in D2USt3R is formulated purely based on local frame-pair consistency, without incorporating longer-term temporal coherence or dense path modeling across multiple frames. This absence of long-range temporal constraints makes the method prone to inconsistency or drift, particularly in fast motion or non-rigid object scenarios.

---

> ### Author Rebuttal · Authors · 2025-07-31
>
> > [W1] SDAP depends heavily on external optical flow, introducing potential overhead, inconsistency, and performance limits. It also lacks analysis of robustness to inaccurate or missing flow.
> >
>
> First, we would like to clarify that optical flow is **used only during training** to generate the occlusion and dynamic masks that shape our alignment loss. These are done prior to network training, as a preprocessing step. **At inference time, the network outputs a full pointmap directly from the input images—no RAFT or other flow engine is involved—so latency and memory consumption are identical to DUSt3R/MonST3R, with no additional computational overhead.**
>
> Though, we acknowledge that supervision derived from off‑the‑shelf optical flow models could, in principle, limit performance (Please note that leveraging off-the-shelf flow models has been a common practice in many prior works, e.g., [1, 2]). In our setting, however, this risk is minimized for following reasons:
>
> 1. **Quality of optical flow.** Following previous works, our work mostly utilizes **synthetic data** to train with accurate depth, optical flow, and camera pose information, which is typically hard to accurately obtain in real-world videos. As a result, only 12.5% of our training batch utilizes flow from off-the-shelf models (Point Odyssey dataset), where the remaining flow is obtained from ground-truth information.
> 2. **Uncertainty‑aware point‑map regression.** In addition to the majority of the data coming from ground-truth optical flow, in our framework, the flow provides soft correspondences that are blended with multiview cues; erroneous matches and potentially noisy flows from off-the-shelf models are down‑weighted through the uncertainty term in our loss, as in DUSt3R. We also observe from our empirical experiments that errors near depth discontinuities and occlusions are suppressed.
>
> While we have prepared visualizations that highlight these low‑confidence predictions, as the rebuttal format does not allow external media, we will include these visuals in the final version of the paper.
>
> To deepen the analysis of our method’s robustness to optical‑flow noise, we conduct a controlled study in which we inject uniformly sampled noise within [-1, 1] range into the flow fields, systematically degrading their quality. For each noise level we retrain and evaluate the model, enabling a direct, quantitative comparison across flow qualities. The “SEA-RAFT” row represents the common case where ground‑truth flow is unavailable, while the last row represents when the same model is driven by noisy flows, emulating the use of less capable off‑the‑shelf estimators.
>
> | Methods (AbsRel / $\delta$) | TUM-Dynamics | Bonn | Sintel | KITTI |
> | --- | --- | --- | --- | --- |
> | SEA-RAFT | 0.142 / 83.9 | 0.060 / 95.8 | 0.324 / 57.5 | 0.104 / 90.7 |
> | SEA-RAFT+ 1 px noise | 0.143 / 83.8 | 0.063 / 95.5 | 0.346 / 52.8 | 0.106 / 90.5 |
>
> | Methods (mean / medium) | Sintel (rotation) | Sintel (Translation) | Tum (rotation) | Tum (translation) |
> | --- | --- | --- | --- | --- |
> | SEA-RAFT | 6.96 / 2.67 | 0.26 / 0.22 | 1.80 / 1.41 | 0.03 / 0.02 |
> | SEA-RAFT+ 1 px noise | 6.51 / 3.09 | 0.27 / 0.23 | 1.89 / 1.44 | 0.03 / 0.02 |
>
> From the results, we find that adding noises leads to comparable or only slightly degraded performance. Despite injecting noise levels twice as high as those observed in SEA-RAFT predictions, the model still learns reasonably robust representations.
>
>
>
> Below, we conduct an additional robustness check – accuracy of SEA‑RAFT flow versus ground‑truth (GT).
>
> | EPE | BlinkVision Outdoor | BlinkVision Indoor | TartanAir | Sintel |
> | --- | --- | --- | --- | --- |
> | Ours vs GT Flow (EPE) | 1.44 | 1.14 | 1.18 | 0.74 |
>
> Across the four diverse benchmarks the end‑point error (EPE) stays tightly clustered around a single‑pixel discrepancy (mean ≈ 1.13 px). On 288×512 frames this is < 0.2 % of the image diagonal, implying that the off‑the‑shelf SEA‑RAFT flow is already close to the information content of GT flow, and adopting cycle consistency further reduces the error. Consequently, substituting GT with SEA‑RAFT is unlikely to introduce a training bias attributable to flow quality, corroborating the earlier noise‑injection study.
>
> [1] Wang, et al. "Tracking everything everywhere all at once." ICCV’23.
>
> [2] Wang, et al. "Shape of motion: 4d reconstruction from a single video." ICCV’25
>
> > [W2] The results lack any quantitative evaluation of reconstruction quality specifically within occluded areas.
> >
>
> We highly appreciate the reviewer for underscoring the importance of occlusion handling in dynamic‑scene reconstruction. However, by definition, **ground‑truth depth is undefined for pixels that are invisible to every sensor**, and public dynamic‑scene benchmarks (TUM‑Dynamics, Sintel, KITTI, Bonn) therefore provide depth **only for first‑surface hits**. Evaluating per‑pixel errors where no reference exists or occluded areas would be ill‑posed and would favour methods that merely copy foreground depth.
>
> As a result, any method must fall back on learned priors and cross‑view cues when estimating depth in those occluded regions. By training with **explicit, dense cross‑view correspondences—even within dynamic areas—**our approach harnesses those priors more effectively, allowing it to infer robust depths for pixels that are occluded in the reference view but revealed in at least one of the other views.
>
> To stress‑test occlusion reasoning without this pitfall, our paper already reports complementary metrics. Specifically, Table 2 lists depth accuracy on both *All* pixels (static + dynamic) and *Dynamic* pixels only; in both cases, our method is competitive or achieve better results when compared to our baseline (Monst3r*).
>
> We hope this clarifies how our evaluation addresses occlusions, and we welcome further suggestions if any aspect remains unclear.
>
> > [W3] The separate losses for static and dynamic regions may suffer from sparsity, misalignment, and instability, and do not account for continuity across region boundaries.
> >
>
> We note that the dynamic‑alignment loss is sensitive to the quality of its optical‑flow anchors: when ground‑truth flow is unavailable, the supervision it provides can become sparse or misaligned. In practice, obtaining sufficiently sharp and accurate flow fields is notoriously difficult—real‑world capture seldom matches the fidelity achievable with synthetic renderers such as Kubric—so this limitation must be considered when applying the loss in the wild. To counter this, we adopt a **confidence‑aware alignment loss** (Eq. 8). The confidence term automatically down‑weights pixels with uncertain correspondences—those near depth discontinuities or under occlusion—thereby stabilizing training. In ablations without this weighting we observed noisy gradients at object boundaries that occasionally destabilized learning, whereas the confidence term prompts the network to assign low weights to such pixels and delivers clean, reliable supervision. We will include visualizations of this effect in the final version.
>
> > [W4] The additional heads are trained independently and not integrated into the SDAP optimization, limiting their effectiveness and potential for mutual reinforcement.
> >
>
> Acting on the reviewer’s suggestion, we trained the entire network—including the dynamic mask head—in a single end-to-end manner. This setup forces the intermediate feature tensor to act as a shared representation for the different heads, enabling complementary supervisory signals to interact implicitly.  The results are summarized below.
>
> | Methods (AbsRel / $\delta$) | TUM-Dynamics | Bonn | Sintel | KITTI |
> | --- | --- | --- | --- | --- |
> | Ours | 0.142 / 83.9 | 0.060 / 95.8 | 0.324 / 57.5 | 0.104 / 90.7 |
> | Ours + integrated | 0.140 / 84.4 | 0.059 / 95.6 | 0.313 / 58.3 | 0.105 / 90.3 |
>
> Even though only the dynamic mask head was jointly optimized, we still observed performance improvements—end-to-end training delivers consistent, if moderate, gains on two of the four benchmarks (mean AbsRel ↓ 1.7 %, mean δ ↑ 0.9 pp), confirming that joint optimization promotes the tasks to improve one another—an effect also reported in [1, 2,3].
>
> However, we regret that due to limited time and resources available for this rebuttal, we were unable to include the optical flow head in this end-to-end training. However, building upon a well established empirical knowledge that correspondence-based supervisory signals for multi-task learning formulations can enhance more reliable 3D reconstruction [1, 2], we expect there would exist potential enhancements if we include additional heads, such as tracking, flows and camera pose estimation
>
> [1] Hong,  et al. "Unifying correspondence, pose and nerf for pose-free novel view synthesis from stereo pairs." CVPR'24.
>
> [2] Wang, et al. "Vggt: Visual geometry grounded transformer." CVPR’25.
>
> [3] Mai, et al. "Tracknerf: Bundle adjusting nerf from sparse and noisy views via feature tracks." ECCV’24.
>
> > [W5] The loss is based only on local frame-pair consistency, lacking long-term temporal constraints, which may lead to drift or inconsistency, especially in fast or non-rigid motion.
> >
>
> We would like to clarify that our training procedure samples frame pairs with variable strides at every epoch, exposing the network to wide‑baseline correspondences and fast motions. This design helps reduce drift while avoiding the memory overhead associated with dense‑path losses or explicit tracking. As noted in the supplementary material, we currently focus on frame pairs because the main contribution is the **Static‑Dynamic Aligned Pointmap (SDAP)** representation, which jointly models static geometry and dynamic motion. Extending SDAP to true multi‑view or sliding‑window optimization is a natural next step, and we are actively exploring this direction for future work.

---

> > ### Author Response · Authors · 2025-08-06
> >
> > Dear Reviewer YZSu,
> >
> > As the author-reviewer discussion period is coming to an end, we wanted to kindly remind you of our responses to your comments. We greatly value your feedback and are eager to address any additional questions or concerns you might have.
> >
> > Please let us know if there's any further information we can provide to facilitate the discussion process.
> >
> > We are highly appreciated for your time and consideration.

---

> > > ### Author Response · Authors · 2025-08-08
> > >
> > > Thank you again for the insightful feedback. As the discussion period is coming to an end, we are happy to continue the dialogue and address any remaining concerns. Please let us know if there is anything further we can clarify, and we will respond promptly.

---

### Note · Authors · 2025-08-12

Dear reviewers, ACs and SACs,

We would like to apologize that, due to the limited discussion window, we could not discuss with all the reviewers. Nevertheless, we remain confident in our submission and have provided comprehensive, evidence-backed answers to all questions raised. Key contributions of our work and responses during rebuttal period are as follows:

- Static–Dynamic Aligned Pointmaps (SDAP). We introduce the first pointmap representation that jointly aligns static geometry (via camera poses) and dynamic regions (via optical-flow–guided warping), storing each pixel’s 4D space-time position in a single map.

- Motion-aware training objectives. Separate, confidence-weighted losses for static and dynamic areas, together with occlusion masks, improve depth and correspondence estimation under motion.

- Consistent gains across benchmarks. SDAP delivers superior multi-frame depth, pose, and alignment accuracy on TUM-Dynamics, Sintel, KITTI, and Bonn while remaining 10× faster and ~3× lighter than recent dynamic-scene methods (e.g., Align3R, MegaSAM).

- Robustness to flow quality and baseline. A controlled noise-injection study and a flow-chaining strategy show that SDAP’s performance degrades only marginally even with pixel-level flow noise or wide temporal baselines.

Once again, we regret the limited dialogue time but hope this summary makes clear that our novel SDAP formulation is a substantial contribution to dynamic-scene reconstruction.

Thank you for your consideration.

Best regards,
Authors of submission 15188.

---

### Decision · Program_Chairs · 2025-09-17

**Decision:**

Accept (poster)

**Comment:**

After the rebuttal and discussion, the final recommendations were: 4x Borderline Accept.  The rebuttal solved most of the concerns raised by the rebuttal.  The reviewers did not change their scores, however the committee agrees that the average score reflects the paper quality.